# Insufficient social distancing may contribute to COVID-19 outbreak: The case of Ijuí city in Brazil

Thiago Gomes Heck[1,2,3,4]* , Rafael Z. Frantz[4,5] , Matias Nunes Frizzo[1,2,3], Carlos Henrique Ramires François[3], Mirna Stela Ludwig[1,2,3], Marilia Arndt Mesenburg[6,7], Giovano Pereira Buratti[2], Lígia Beatriz Bento Franz[2,8], Evelise Moraes Berlezi[2,3,8]

**1** Research Group in Physiology, Regional University of Northwestern Rio Grande do Sul State, Ijuí, Rio Grande do Sul State, Brazil, **2** Postgraduate Program in Integral Attention to Health, Regional University of Northwestern Rio Grande do Sul State, Ijuí, Rio Grande do Sul State, Brazil, **3** Medicine Course, Department of Life Sciences, Regional University of Northwestern Rio Grande do Sul State, Ijuí, Rio Grande do Sul State, Brazil, **4** Postgraduate Program in Mathematical and Computational Modeling, Ijuí, RS, Brazil, **5** Department of Exact Sciences and Engineering, Regional University of Northwestern Rio Grande do Sul State, Ijuí, Rio Grande do Sul State, Brazil, **6** Federal University of Pelotas (UFPel), Pelotas, Rio Grande do Sul State, Brazil, **7** Federal University of Health Sciences of Porto Alegre (UFCSPA), Porto Alegre, Rio Grande do Sul State, Brazil, **8** Research Group in Human Aging, Regional University of Northwestern Rio Grande do Sul State, Ijuí, Rio Grande do Sul State, Brazil

☯ These authors contributed equally to this work.
* thiago.heck@unijui.edu.br

**Data Availability Statement:** We used public data from government sites: Brazilian national government COVID-19 website. Available in https://covid.saude.gov.br/. Rio Grande do Sul State

## Abstract

The coronavirus disease that emerged in 2019 (COVID-19) is highly contagious and has given way to a global pandemic. A present COVID-19 has high transmission rates worldwide, including in small Brazilian cities such as Ijuí. Located in the northwest part of the state of Rio Grande do Sul (RS) and with a population of 83,475, Ijuí was selected as the site of a population-based survey involving 2,222 subjects, from April to June 2020. Subjects were tested for the presence of antibodies against coronavirus (SARS-CoV-2) and answered questions regarding social distance adherence (SDA), daily preventive routines (DPR), comorbidities, and sociodemographic characteristics. In parallel, the local government registered the official COVID-19 cases in Ijuí, as well as the mobile social distancing index (MSDI). In this study, we demonstrate that there was a decrease in the levels of SDA, DPR and MSDI before the beginning of COVID-19 community transmission in Ijuí. Furthermore, we provide predictions for the number of COVID-19 cases, hospitalizations, and deaths in the city. We conclude that insufficient social distancing, as evidenced by different methods, may be related to the rapid increase of COVID-19 cases in Ijuí. Our study predicts an approaching outbreak of COVID-19 in Ijuí through community spread, which could be avoided or attenuated with increased levels of social distancing among the population.

## Introduction

At the end of 2019, some cases of "pneumonia of unknown cause" were noticed by the Wuhan Municipal Health Commission in China's Hubei Province [1]. Collected bronchoalveolar-

government COVID-19 website. Available at https://covid.saude.rs.gov.br/.

**Funding:** The author(s) received no specific funding for this work.

**Competing interests:** The authors have declared that no competing interests exist.

**Abbreviations:** COVID-19, Coronavirus Disease emerged in 2019; DPR, Daily Preventive Daily; MSDI, Mobile Social Distancing Index; RS, State of Rio Grande do Sul; RT-PCR, Real-Time Reverse Transcription–Polymerase Chain Reaction; SDA, Social Distancing Adherence; WHO, Word Health Organization.

lavage samples were consistent with an RNA virus of the *Coronaviridae* family [2]. Thereafter, the World Health Organization (WHO) named the novel infectious pneumonia "coronavirus disease 2019" or COVID-19 [3]. COVID-19 was quickly proven to be highly contagious, reaching approximately 300 cases in China by January 2020 [4]. Soon COVID-19 became a pandemic, with more than 44,000 infections and more than 1,000 deaths in China, with 441 cases outside China in 24 countries [5].

The first case of COVID-19 in Brazil was reported on February 27, 2020 in the city of São Paulo. Based on published events, eight of the 27 federated units of Brazil present cumulative mortality rates above 10 per 100,000 inhabitants: four in the north, two in the northeast, and two in the southeast region (including Rio de Janeiro and São Paulo) [6]. Until November 2020, Brazil officially recorded 5,468,270 cases of COVID-19 (2,602 per 100,000 inhabitants) and 158,456 COVID-19 deaths (75 per 100,000 inhabitants). The five federative units with the highest mortality counts are São Paulo (39,007 deaths), Rio de Janeiro (20,376 deaths), Ceará (9,325 deaths), Minas Gerais (8,872 deaths), and Pernambuco (8,587 deaths). The highest cumulative mortality rates above 10 per 100,000 inhabitants are found in Ceará, with 102 deaths per 100,000 inhabitants [7]. In the state of Rio Grande do Sul (RS), the southernmost state in Brazil with 11.3 million people, the first case of COVID-19 was diagnosed on February 29, 2020. As of August 6, 2020, 76,563 confirmed cases (673 per 100,000 inhabitants) and 2,163 deaths (19 per 100,000 inhabitants, 2.8% of confirmed cases) have been reported [8, 9]. As of November 2020, RS recorded 240,694 COVID-19 cases (2,116 per 100,000 inhabitants) and 5,699 deaths (50 per 100,000 inhabitants) [8].

Although a significant investment has been made worldwide to provide antiviral prophylaxis for COVID-19, to test different drugs for prevention or treatment COVID-19 cases, and to develop vaccines [10], current recommendations to reduce the spread of COVID-19 include physical distancing [11], quarantining, and large-scale lockdowns of entire populations [12, 13]. Evidence indicates that the implementation of social distancing can suppress COVID-19 transmission rates to prevent the disease from overwhelming the healthcare system. In an analysis of 49 countries, Atalan [14] showed that the COVID-19 pandemic can be suppressed by lockdown measures. In another study including data from 131 countries [15], a decrease in the transmission rate of COVID-19 was observed within 1–3 weeks following the introduction of school closures, workplace closures, public events bans, stay-at-home orders, and limits on internal movement. However, the reduction of transmission ranged from 3% to 24% approximately one month following the introduction of the recommendations, and the effect was only statistically significant for public events bans [15]. Similarly, in New Zealand (a country of 4.886 million inhabitants), the estimated COVID-19 case infection rate decreased from 8.5 to 3.2 per one million people after the implementation of a nationwide the lockdown, resulting in a low relative burden of disease [16]; until now New Zealand has accumulated only 1,973 COVID-19 cases and 25 deaths. Although social distancing and lockdown measures appear to be successful, there is "social fatigue" associated with following these recommendations, leading many societies to return to a usual lifes, increasing COVID-19 transmission [17, 18].

To investigate this, a population-based survey EPICOVID-RS was conducted in nine cities in Rio Grande do Sul from April to June of 2020 [9], including the city of Ijuí, population 83,475, located in the northwestern region of the state. The survey tested 2,500 subjects for the presence of antibodies against SARS-CoV-2 over the course of five different rounds of ~500 subjects each. The seroprevalence of antibodies against SARS-CoV-2 was undetectable (i.e. 0 positive cases) on April 12[th] (round 1), and increased to 0.042% [8, 9]. Participants answered questions regarding social distancing adherence (SDA) and daily preventive routines (DPR) [9]. In parallel, the local government registered and published the official number of COVID-19 cases in Ijuí (for details, see the official government website [19]), while the mobile social

distancing index (MSDI) in Ijuí was monitored by means of data collected from mobile geolocation [20].

With no pharmaceutical treatment for COVID-19 available, and a vaccine still months away, interventions have focused on quarantine and social distancing. The aim of these strategies is to slow down the spread of infection and reduce the intensity of the transmission (i.e., "flatten the curve") [21]. The initial COVID-19 outbreak in Wuhan, China indicates that critical care capacity can be exceeded if distancing measures are not implemented quickly or strongly enough [22]. An effective social distancing strategy and monitoring may reduce the risk of overwhelming the health system, allowing adequate patient care and decreasing mortality rates [22]. Using data on the local COVID-19 cases and social distancing behavior, we demonstrate that low levels of social distancing were insufficient to prevent the COVID-19 outbreak in Ijuí, Brazil.

## Materials and methods

### Study design, subjects, and ethics

Ijuí (28˚23'16 S and 53˚54'53" W) is the most populous city in the northwest region of Rio Grande do Sul. With 83,475 residents, it is considered a city of students ("university city") and a center of hospital and university resources. Furthermore, it is the largest and most important population center in the region, with a population rounding 150,000 people. Ijuí has a high Human Development Index (HDI) score of 0.781, above the overall HDI of Brazil (0.761). Ijuí has a high score for all three parameters measured for HDI calculation: education (HDI-E = 0.707), with 98.9% of children aged 6–14 in school; longevity (HDI-L = 0.858), with an average life expectancy of 76.48 years; and per capita income (HDI-R = 0.786), with R$ 38,341.14 (approximately $7,119.33 per capita/year [23].

This study uses an interdisciplinary approach to investigate whether the behavior of citizens in Ijuí is associated with an increase in the number of COVID-19 cases. First, we considered official data about COVID-19 cases, which is updated daily on the government website and plotted as a cumulative case line graph showing the date of the initial symptoms and the date of the confirmed test [19]. Second, we analyzed the MSDI between February 1, 2020 and July 5, 2020 [20]. Third, we analyzed data from the study EPICOVID-RS in Ijuí regarding SDA and DPR [9]. Finally, we make predictions about the COVID-19 pandemic in Ijuí regarding the number of cases, hospitalizations, and deaths projected under different scenarios of transmission rates (starting after the 100[th] case), compared to actual COVID-19 data.

The National Institute of Geography and Statistics divides the state of RS [23] into eight intermediary regions, and the main city in each region was selected for the EPICOVID-RS study [9]. Ijuí is one of the main cities, so it was selected for this study. Ethical approval was obtained from the Brazilian National Ethics Committee (process number 30415520.2.0000.5313), and all participants provided written informed consent.

### Official COVID-19 data in Ijuí

The COVID-19 Municipal Scientific Committee of Ijuí developed an information panel with public data from the Municipal Epidemiological Surveillance that provides information on the number of confirmed COVID-19 cases, hospitalizations, recoveries, and deaths [19]. The number of confirmed cases is also listed by age and gender. Confirmed cases consist of data received from the laboratory analysis, including real-time reverse transcription–polymerase chain reaction (RT-PCR), rapid tests, and chemiluminescence tests. The numbers of confirmed cases published in the bulletins of the municipal health department may differ from those reported by the state information system (e-SUS VE) due to the frequency of updating

the system. Furthermore, the number of confirmed cases reported by hospitals in Ijuí may differ from municipal data because of the origin of hospitalized patients, which are from different neighboring cities. Our study takes into consideration only data from the Ijuí Municipal Health Department for people residing in Ijuí.

## Mobile Social Distancing Index (MSDI)

The MSDI was created by the Brazilian private company Inloco (www.inloco.com.br), which has developed a new geolocalization technology that creates a unique identity for users of mobile devices [20]. The company currently monitors a stable set of 60 million mobile devices across Brazil. Each device stores and sends anonymous location data every time it connects to the Internet. The MSDI is computed on a daily basis and reflects the percentage of mobile devices in a given municipality that remain within a radius of 450 meters from the point identified as home; the system is precise to within <3 meters [20]. Within the period considered in this study (from February 1, 2020 to July 5, 2020), the dataset had a total of approximately 1 million records for every city in Brazil. From this dataset we extracted daily MSDI measurements for the city of Ijuí.

## Population-based survey protocol: EPICOVID-RS in Ijuí

Multistage sampling based on census listings updated in 2019 was used to select ten households at random within each census tract in Ijuí. All household members were listed at the beginning of the visit, and one individual was randomly selected through an app used for data collection. The survey rounds took place in the following periods: April 11–13 and 25–27, May 9–11 and 23–25, and June 27–29 (for data presentation in figures, we considered the mean value for the 3-day interval). If there were any refusals at the household level, the next household on the list was selected until a total of ten families had been sampled. In the second wave of data collection, field workers went to the house visited in the first wave and then selected the tenth household to its right. The same procedure was performed for the third, fourth, and fifth waves. In case of refusal, the next house to the right side was selected. In the case of acceptance at the household level, but the selected individual refused to provide a sample, a second household member was selected. If this person also refused, the field workers moved on to the next household on the list. (for more details, see reference [9]). Prevalence of antibodies against SARS-CoV-2 was assessed with the WONDFO SARS-CoV-2 Antibody Test (Wondfo Biotech Co., Guangzhou, China) using finger-prick blood samples. The sensitivity and specificity of this rapid test have been previously validated [9, 24].

Additionally, at each study wave participants answered short questionnaires, including sociodemographic information (sex, age, medical history, schooling, and race), COVID-19-related symptoms, use of health services, compliance with social distancing measures, and use of face masks. The questions on social distancing were as follows: 1) "To what extent are you managing to follow the social distancing guidance from the health authorities, i.e., staying at home and avoiding contact with others?" This was scored on a five-point scale, with the following alternatives read aloud to the respondent: "very little," "little," "some," "quite," and "practically isolated from everyone;" 2) "What have your routine activities been?" The alternatives were: "staying home all the time," "only leaving home only for essentials, such as groceries," "leaving home from time to time to run errands and stretch legs," "going out every day for regular activities," and "out of the house all day, every day, either for work or for other regular activities." This questionnaire passed an internal validation before it was applied in this study. After this, the questionnaire was applied in 133 cities covering all regions of Brazil [6, 24–26]. The dataset used to produce the analyses presented in this study is freely available at

http://www.rs.epicovid19brasil.org/banco-de-dados/ and from the corresponding author upon request. The questionnaire is available in the S1 File.

Field workers used tablets or smartphones to make a full audio recording of each interview and to register all answers and photograph the test results. All positive or inconclusive tests were read by a second observer, as were 20% of the negative tests to validate the results. If the selected subject in a household had a positive result, all other family members were invited to be tested. One day before to interviewing study participants, field workers were tested and found to be negative for COVID-19. Field workers were also provided with personal protection equipment (face shield, masks, gloves and disposable lab coats) that was discarded after visiting each home (except the face shield that was cleaned with alcohol 70%). Positive COVID-19 cases were reported to the municipal and statewide COVID-19 surveillance system. The number of COVID-19 positive subjects was then noted in social media by the RS government three days after each wave was completed.

## Prediction of COVID-19 cases, hospitalizations, and deaths

Our analyses focused on declared social distancing behavior and daily behavior characteristics, as well as the association between comorbidities and social distancing behavior using logistic regression. Data were reported as odds ratios (OR) with an upper and lower limit of the confidence interval (95%CI). Estimation of new COVID-19 cases was first performed using city government data on COVID-19 from March 22, 2020 to May, 2020 (nine points). Predictions were calculated using equations generated as follows: exponential, defined as proportional rate growth using the expression $y = Y_0 - (V_0 / K) \times (1 - e^{-kx})$, or linear, defined as a straight line using the expression $y = mx + c$. After curve fitting, correlations were predicted (extrapolated) using the online version of MyCurveFit software (https://mycurvefit.com/), and the results were plotted using GraphPad 8.0 (GraphPad Software, San Diego, CA).

To estimate the number of exposed subjects, COVID-19 cases, hospitalization rates, and deaths after the 100[th] case of COVID-19 in Ijuí, we used the SEIR (Susceptible → Exposed → Infected → Recovered) model [27]. A full description of the equation is provided by Wu and colleagues [28]. We used the following parameters for the predictions: a population of 83,200; 100 initial cases; $R_0 = 2.79$ as the basic reproductive number that represents the number of secondary infections that each subject produces, (based on the median transmission rate documented in 12 previous studies [29, 30]); $R_0 = 1.44$ as the reproduction number estimated for RS on May 8, 2020 [31]; a virus incubation time of 5.21 days; the time that individuals remain transmitting the virus as 2.3 days; a time between incubation and death of 30 days; hospitalization days until recovery as 21 days (severe) and 14 days (mild); time of hospitalization as 14 days; and a hospitalization rate of 12% and a mortality of 2.3% (based on RS governmental data until the end of June 2020 [8]). We used these parameters to predict the number of exposed subjects, number of COVID-19 cases, hospitalization rate, and deaths after the 100[th] case of COVID-19 in Ijuí under six different situations: 1) maintaining the transmission rate of community spread of COVID-19 as $R_0 = 2.79$; 2) reducing the transmission rate by 50% ($R_t = 1.44$) exactly 30 days after the 100[th] case; 3) reducing the transmission rate by 50% ($R_t = 1.44$) exactly 15 days after the 100[th] case; 4) maintaining the transmission rate as $R_0 = 1.44$; 5) reducing the transmission rate by 50% ($R_t = 0.79$) exactly 30 days after the 100[th] case; and 6) reducing the transmission rate by 50% ($R_t = 0.79$) exactly 15 days after the 100[th] case. These estimations were calculated using a free online tool for epidemic profile calculations from University of São Paulo [32].

## Results

The daily MSDI computed using data from mobile monitoring for the city of Ijuí is shown in Fig 1A. On the first day of mobile monitoring, the MSDI was 30.7% (Saturday February 1, 2020), after which point it increased to 39.3% (Sunday February 2, 2020) and then decreased to 27.5% in the first weekday registered (Monday February 3, 2020). These data describe the normal social behavior in Ijuí, which remained stable with only a few fluctuations until mid-March, when MSDI grew to 70.2% (Sunday March 22, 2020) and stabilized between 50% and 55% on subsequent weekdays. However, at the end of June, the MSDI decreased to 31.4% (June 19, 2020), reaching the lowest value since city's first registered case of COVID-19 on March 18, 2020 (Fig 1B). The last registered MSDI value was 35.3% (June 30, 2020). For a fair comparison, percentages pointed above the open circles in Fig 1A correspond to MSDI on weekends, whereas the closed circles correspond to MSDI on working days. We observed an increase in distancing on the weekends, indicating that more people tended to stay home on weekends vs. weekdays. Specifically, on weekends MDSI was ~12% higher than on working days (44.87 ± 9.70 [95% CI: 41.99–47.75] vs. 36.07 ± 7.45 [95% CI: 34.66–37.48], $p < 0.0001$, Student's t-test). However, a decrease in overall MSDI (weekends and weekdays) was registered between March 22, 2020 (70.2%) and June 21, 2020 (48.2%). From a macro point of view, it is evident that the population of Ijuí is loosening social distancing and increasing social interaction up to the end of June.

The person to test positive for COVID-19 in Ijuí (Fig 1B) started to feel symptomatic on March 18, 2020 and was diagnosed shortly thereafter. Until May 17, 2020, there was a slight increase in the number of cases registered in Ijuí (Fig 1B). However, on June 7, 2020 there was a 2.1-fold increase in total COVID-19 cases. A majority of these cases occurred in women (52.4%) and people younger than 60 years of age (86%) (Fig 1C). Until the end of June, Ijuí had a 9.5% hospitalization rate for positive COVID-19 cases; 74.1% of hospitalizations occurred in people over 50 years of age, and no hospitalizations occurred for people under 30 years of age. The majority (70.4%, n = 19) of the hospitalized cases were men as opposed to women (29.6%, n = 8). Until the end of June, there was just one death caused by COVID-19 in Ijuí (an 85-year-old oncologic patient with diabetes and hypertension, male). The urban distribution of COVID-19 cases in Ijuí indicates that approximately 30% were downtown inhabitants (S1 Fig in S1 File).

In each round of the population-based survey, >400 adults were surveyed and tested for the presence of SARS-CoV-2 antibodies, for a total of 2,222 study participants. Characteristics of the study population are described in Tables 1 and 2. The majority of the subjects interviewed were women (~60%) and white (~80%), with a roughly equal distribution across age and education categories (Table 1). Between 32% and 39% of participants reported having hypertension and ~13% and ~10% had diabetes and asthma, respectively (Table 2).

The social distancing population-based survey (initiated on April 12, 2020) revealed that the majority of the subjects reported high SDA (Fig 2A). However, we observed a decrease in the proportion of subjects that declared high adherence in parallel to an increase in the proportion that reported partial adherence (Fig 2A). Similarly, the proportion of the subjects that reported staying at home all the time or only going out for essential needs has decreased, while there was an increase in the number of people that declared leaving their homes daily for work (Fig 2B). This is consistent with the decrease in MSDI just 15 days after the first survey wave, which continued to decrease until the end of May (blue lines in Fig 2C). In contrast, there was a significant increase in the number of COVID-19 cases in Ijuí during the period of May to June (red line, Fig 2C), which may evocate a recovered of SDA and DPR approximately one month later (June) (blue lines in Fig 2C).

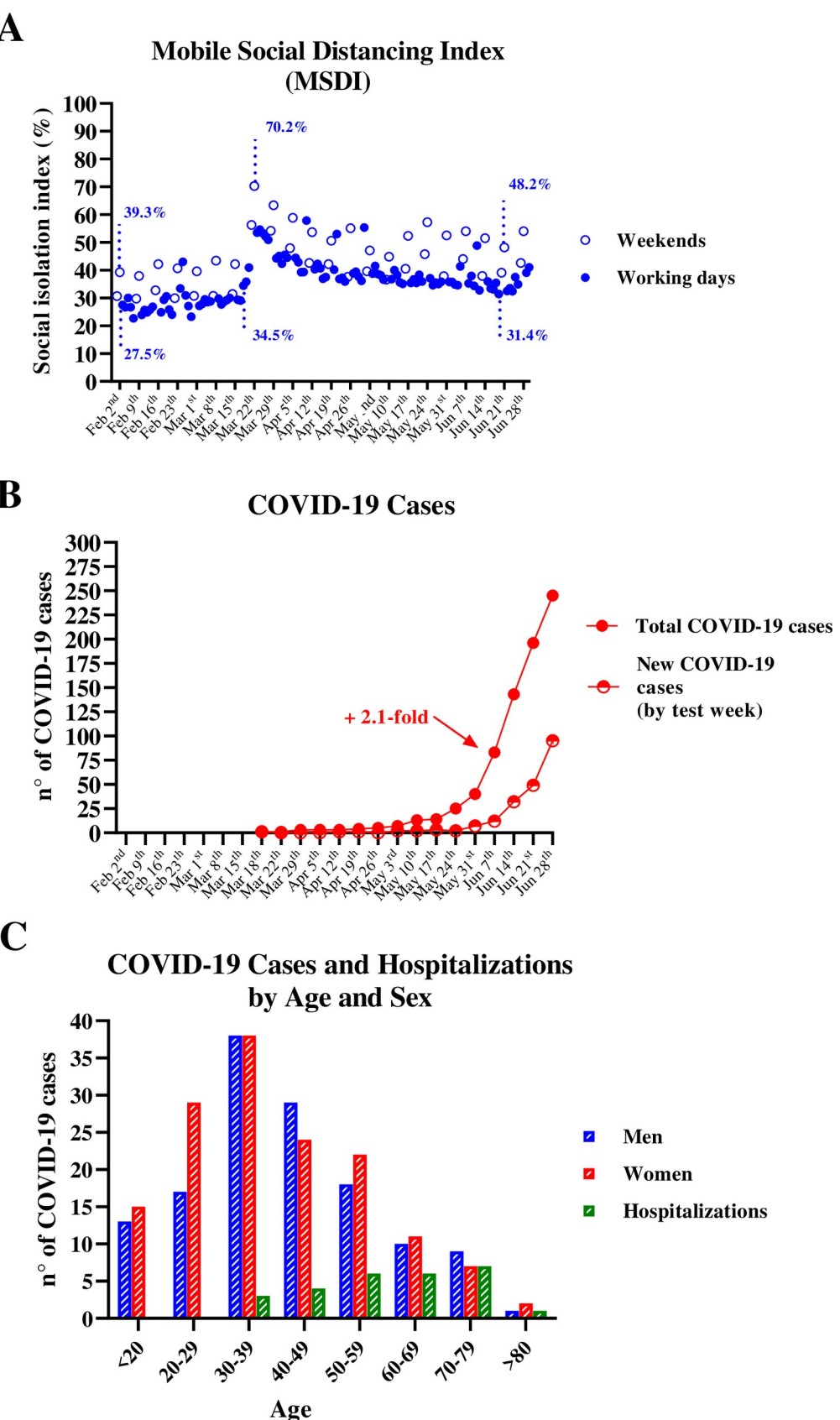

**Fig 1. Mobile social distancing index and COVID-19 cases in Ijuí, Brazil. A)** Mobile Social distancing index (MSDI) **B)** Total and new weekly COVID-19 cases **C)** Distribution of COVID-19 cases and hospitalizations by age.

We have also analyzed two scenarios to estimate the number of cases in Ijuí after May 17, 2020 (Fig 3A): First, a sustained controlled community infection scenario represented by a linear trend (green line in Fig 3A, estimated by the equation $y = 1.416667x - 1.416667$, $R^2 = 0.848$, $P = 0.0004$) and second, an uncontrolled outbreak community infection in Ijuí, represented by an exponential progression in the number of cases (blue line in Fig 3B, estimated by the equation $y = 1.351649 - (-0.2424262/-0.3155474) \times (1 - e^{\wedge(+0.3155474x)}$, $R^2 = 0.955$, $P = 0.00019$). Worryingly, the number of actual COVID-19 cases registered through July 7, 2020 (277, red line in Fig 3A) was higher than what was estimated by linear and exponential prediction (23 and 120, respectively). The exponential estimation (blue line) predicted that Ijuí could reach 1,000 cases of COVID-19 on August 23, 2020, indicating that Ijuí's public health system could be overwhelmed.

Next, we analyzed predictions for the months after the 100th case of COVID-19 in Ijuí. In the worst scenario ($R_0 = 2.79$), the public and private healthcare systems would be overwhelmed with >4,000 hospitalizations and >1,500 deaths in 120 days (Fig 3B). In contrast, if the transmission rate of COVID-19 is reduced by 50% 30 days after the 100th case ($R_t = 1.40$), a moderate decrease in the impact on the healthcare system is expected (Fig 3C). We further analyzed the impact on the transmission rate of COVID-19 if the transmission rate fell by 50% ($R_t = 1.40$) 15 days after the 100th case (Fig 3D). This scenario reduced the peak infection to 1,314 cases, expected to occur in 66 days. Consequently, the peak of hospitalization (1,582) would occur later, 85 days after the 100th case, and result in 862 deaths in 120 days. If this RS

**Table 1. Sociodemographic characteristics of Ijuí subjects by date of population-based survey.**

| | Survey 1 (April 11–13th) | Survey 2 (April 25–27th) | Survey 3 (May 9–11th) | Survey 4 (May 23–25th) | Survey 5 (Jun 27–29th) |
|---|---|---|---|---|---|
| | N = 420 | N = 426 | N = 454 | N = 450 | N = 472 |
| **Sex** | N (%) | N (%) | N (%) | N (%) | N (%) |
| Men | 167 (39.8) | 165 (38.7) | 174 (38.3) | 198 (44.0) | 185 (39.3) |
| Women | 253 (60.2) | 261 (61.3) | 280 (61.7) | 252 (56.0) | 287 (60.7) |
| **Age** | | | | | |
| 20–29 | 73 (17.4) | 53 (12.4) | 61 (13.4) | 66 (14.7) | 54 (11.4) |
| 30–39 | 65 (15.5) | 66 (15.5) | 61 (31.4) | 83 (18.4) | 72 (15.3) |
| 40–49 | 69 (16.4) | 67 (15.7) | 81 (17.8) | 77 (17.1) | 88 (18.6) |
| 50–59 | 61 (14.5) | 88 (20.7) | 88 (19.4) | 80 (17.8) | 85 (18.0) |
| 60–69 | 80 (19.1) | 75 (17.6) | 73 (16.1) | 79 (17.6) | 102 (21.7) |
| 70–79 | 46 (11.0) | 60 (14.1) | 64 (14.1) | 47 (10.4) | 51 (10.8) |
| 80+ | 26 (6.2) | 17 (4.0) | 26 (5.7) | 18 (4.0) | 20 (4.2) |
| **Educational attainment** | | | | | |
| None/primary school | 131 (31.2) | 7 (1.6) | 13 (2.9) | 6 (1.3) | 12 (2.5) |
| Lower secondary school (complete and incomplete) | 66 (15.7) | 149 (35.0) | 152 (33.5) | 145 (32.2) | 141 (29.9) |
| Upper secondary school (complete and incomplete) | 144 (34.3) | 138 (32.4) | 128 (28.2) | 118 (26.2) | 164 (34.8) |
| University (complete and incomplete) | 79 (18.8) | 132 (31.0) | 161 (35.5) | 181 (40.2) | 155 (32.8) |
| **Ethnicity** | | | | | |
| White | 347 (82.8) | 322 (75.8) | 360 (80.4) | 360 (80.4) | 377 (81.2) |
| Others (indigenous. black. asian) | 72 (17.2) | 103 (24.2) | 88 (19.6) | 88 (19.6) | 87 (18.8) |

**Table 2. Comorbidities characteristics of Ijuí subjects by date of population-based survey.**

|  | Survey 1 (April 11–13[th]) | Survey 2 (April 25–27[th]) | Survey 3 (May 9–11[th]) | Survey 4 (May 23–25[th]) | Survey 5 (Jun 27–29[th]) |
|---|---|---|---|---|---|
|  | N = 420 | N = 426 | N = 454 | N = 450 | N = 472 |
|  | N (%) | N (%) | N (%) | N (%) | N (%) |
| Hypertension | 145 (34.5) | 153 (35.9) | 177 (39.0) | 147 (32.7) | 174 (36.9) |
| Diabetes | 54 (12.9) | 53 (12.4) | 60 (13.2) | 60 (13.3) | 59 (12.5) |
| Asthma | 44 (10.5) | 45 (10.6) | 54 (11.9) | 49 (10.9) | 53 (11.2) |
| Cancer | 25 (6.0) | 18 (4.2) | 18 (4.0) | 19 (4.2) | 24 (5.1) |
| Kidney disease | 14 (3.3) | 6 (1.4) | 18 (4.0) | 11 (2.4) | 14 (2.9) |
| Heart disease | 39 (9.3) | 36 (8.5) | 54 (11.9) | 29 (6.4) | 50 (10.6) |
| Other diseases | 51 (12.1) | 46 (10.8) | 53 (11.7) | 67 (14.9) | 54 (11.4) |

transmission rate ($R_0$ = 1.44) persisted without any alteration after the 100[th] case, it would result in a higher number of infections, hospitalizations, and deaths peaking after two months (Fig 3E). However, a 50% decrease in the transmission rate 30 days after the 100[th] case ($R_t$ = 0.79), would considerably minimize the impact of COVID-19 on the healthcare system, resulting in a peak of 156 daily cases 34 days later, a peak of 120 daily hospitalizations 60 days later, and a peak of 62 daily deaths 120 days later (Fig 3F). Finally, if the transmission rate was reduced by 50% ($R_t$ = 0.79) 15 days after the 100[th] case (Fig 3G), the infection peak would reach 75 daily cases and 52 daily hospitalizations, resulting in 27 deaths. The proportion of the whole population that needs recovery decreased from approximately 82.0% in the worst prediction to 1.37% in the best one. Specifically, in the most optimistic prediction (a 50% reduction in the transmission rate 15 days after the 100[th] case of COVID-19) approximately 98.7% of the population of Ijuí would remain without infection, and approximately 1.37% would be recovered (1,162 subjects).

## Discussion

Our study describes the SDA, DPR, and MSDI among the population of Ijuí, Brazil, and predicts the progression of COVID-19 under different scenarios. To the best of our knowledge, this is the first study to describe the social distancing behavior of a community based on a population-survey procedure and mobile monitoring data that preceded the COVID-19 outbreak. Furthermore, this is the first report to estimate the progression of COVID-19 in Ijuí.

Although there are currently many studies about social distancing behavior and COVID-19 cases in Brazil, it is imperative to study local data separately. Brazil is a vast country comprised of 26 federative states and the Federal District, and there are many cultural, economic, educational, and geographic differences between states and between different cities in the same state. Given these differences, trends in the number of cases and deaths differs between states and cities, and municipalities have the autonomy to determine which measures to adopt in order to best mitigate COVID-19 according to their respective scenarios [33]. All of Brazil's states implemented distancing measures, mostly after March 15, 2020. Partial economic lockdown was implemented before the tenth confirmed case of COVID-19 by 18 (67%) states and before the first death from COVID-19 by 24 (89%) of the states [33]. In April 2020, of nine major cities in Rio Grande do Sul, the biggest cities (Porto Alegre and Santa Maria) exhibited the highest degree of social distancing, while Ijuí had a less favorable pattern, with a strikingly higher percentage reported being "out of the house all day" [25].

In February 2020, the city of Ijuí, as well as its neighboring cities, had not registered any cases of COVID-19. The first case registered in the state of RS occurred on March 10, 2020 in

**A**

### Social Distancing Adherence (SDA)

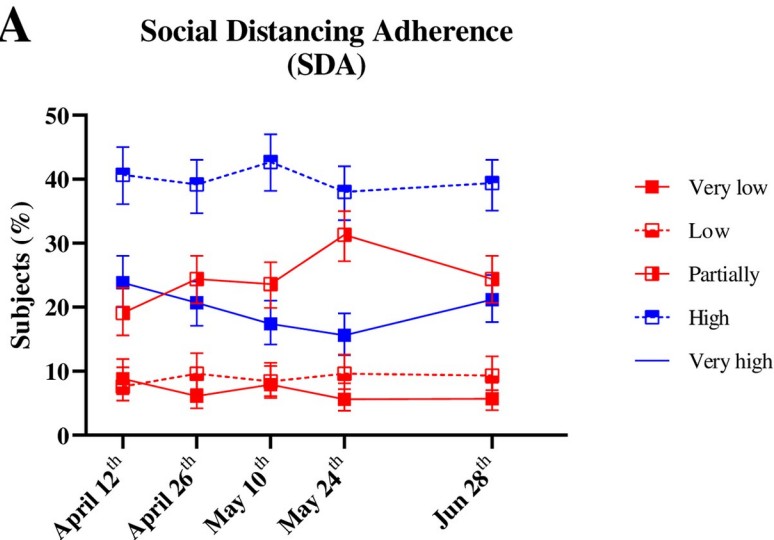

- Very low
- Low
- Partially
- High
- Very high

**B**

### Daily Preventive Routine (DPR)

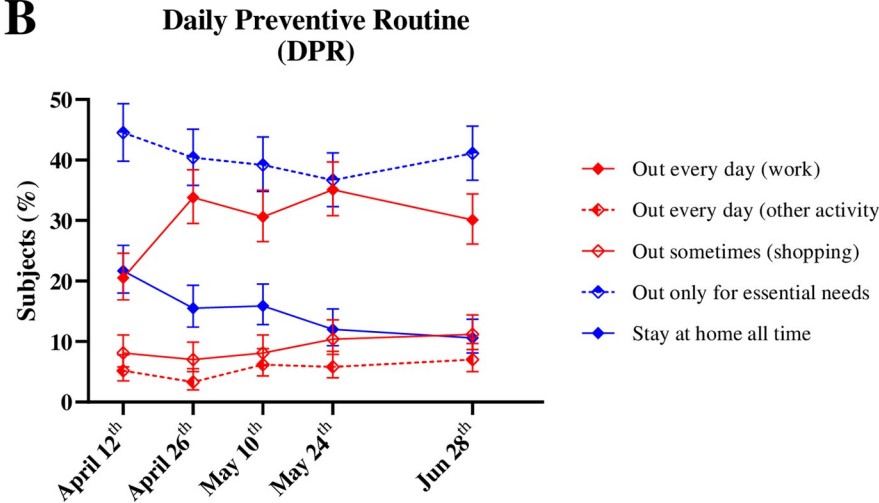

- Out every day (work)
- Out every day (other activity)
- Out sometimes (shopping)
- Out only for essential needs
- Stay at home all time

**C**

### Social distancing behavior vs. COVID-19 Cases

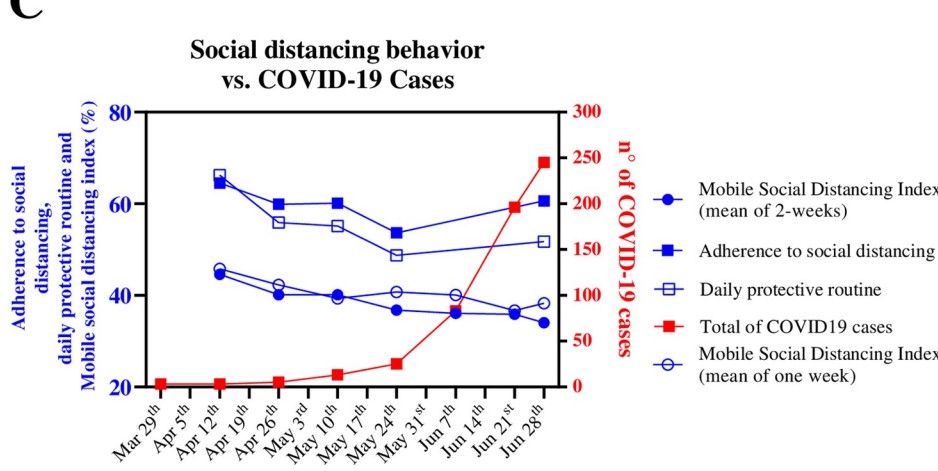

- Mobile Social Distancing Index (mean of 2-weeks)
- Adherence to social distancing
- Daily protective routine
- Total of COVID19 cases
- Mobile Social Distancing Index (mean of one week)

**Fig 2. The association of social distancing adherence (SDA), daily preventive routine (DPR), and mobile social isolation index (MSDI) with COVID-19 cases in Ijuí, Brazil. A)** Social distancing adherence, as indicated in the population-based survey study. **B)** Daily preventive routine, as indicated in the population-based survey study **C)** Association between total COVID-19 cases and measures of social distancing.

the city of Campo Bom– 402 km away from Ijuí, and very close to the capital of the state, Porto Alegre. The first case of COVID-19 registered in Ijuí dates back to March 18, 2020. Ijuí joined the official counting of cases after eight days; most probably, the absence of cases before this date explains the non-change in the percentage of social distancing (respecting its natural peaks on Sundays).

On March 19, 2020 the city administration issued its most restrictive decree in the analyzed period, The decree restricted public transport, closed stores, suspended classes at schools and universities, and established special protocols for restaurants and other services for one week [19]. Accordingly, on March 22, 2020 the MSDI reached its highest value: 70.2%. Other subsequent decrees were issued in Ijuí, but with more relaxed restrictions. In parallel, the Federal Government of Brazil has been minimizing this pandemic and, in most cases, encouraging people to keep their regular routines [34]. Thus, we observe a continuous reduction of the social distancing, to the point that at the end of June, MSDI fell to the same levels seen at the beginning of the analyzed period, when there were no cases of COVID-19 in Ijuí or even in Brazil as a whole [7, 35]. Our results were similar to those reported by Oliveira and colleagues [36], presenting a mean isolation index from February 1, 2020 to April 10, 2020 of 40.2%, ranging from 18.5% to 69.4%. Specifically, this study analyzed mean isolation index in the states of São Paulo (13.5% to 67.9%) and Rio de Janeiro (16.6% to 69.4%) and found that social isolation indexes of 46.7% have the highest accuracy (93.9%) to predict $R(t) < 1$ [36], which means that the epidemic is slowing. These data reinforce the validity and reliability of MSDI as a behavioral indicator of social distancing.

In the final days of June 2020, we observed a slight increase in social distancing, most likely as a consequence of a high number of new cases reported in the second half of June, which encouraged the city administration to review its protocols and implement stricter recommendations [19]. The mean MSDI over the total period of mobile monitoring was 38.5%. Taking into account that "staying at home" was defined as a mobile not moving outside a 450 m radius [20], it is possible that the actual social distancing level in Ijuí was below 30% in regular times, since a 450 m distance still allows neighborhood social interactions, including small markets.

Our results about SDA and DPR may be related to the official recommendations by local and state governments. Since May 10, 2020 state of RS has implemented the "Controlled Social Distancing Model" (CSDM). To minimize the spread of COVID-19, this model outlines the safety of various economic activities in a color-coded manner, ranging from yellow (representing a low risk of infection and promoting high health care capacity) to black (representing a high risk of transmission and promoting high occupation rates of hospitalization and intensive care unit visits). Since May 10, 2020, every city in RS is classified according to this model to represent the progression of the COVID-19 outbreak [8]. Ijuí was categorized as yellow from May 10 to June 8, when it became orange (one level above of risk) because of the rapid increase in the number of cases (95 new cases) and hospitalization rates in the intensive care units (5 subjects) in one week and the subsequent decrease in the ratio of recovered-to-active ratio cases. As a local initiative to counteract the COVID-19 outbreak, Ijuí Municipal Decree number 7.107 (issued June 16, 2020) improved the recommendations to decrease the risk of community spread of the coronavirus [19]. In detail, the decree allows the maintenance of regular service in restaurants (between 7 AM and 11 PM) with reduced capacity, while snack bars and coffee shops were allowed to work only in delivery or drive-thru settings. The non-essential

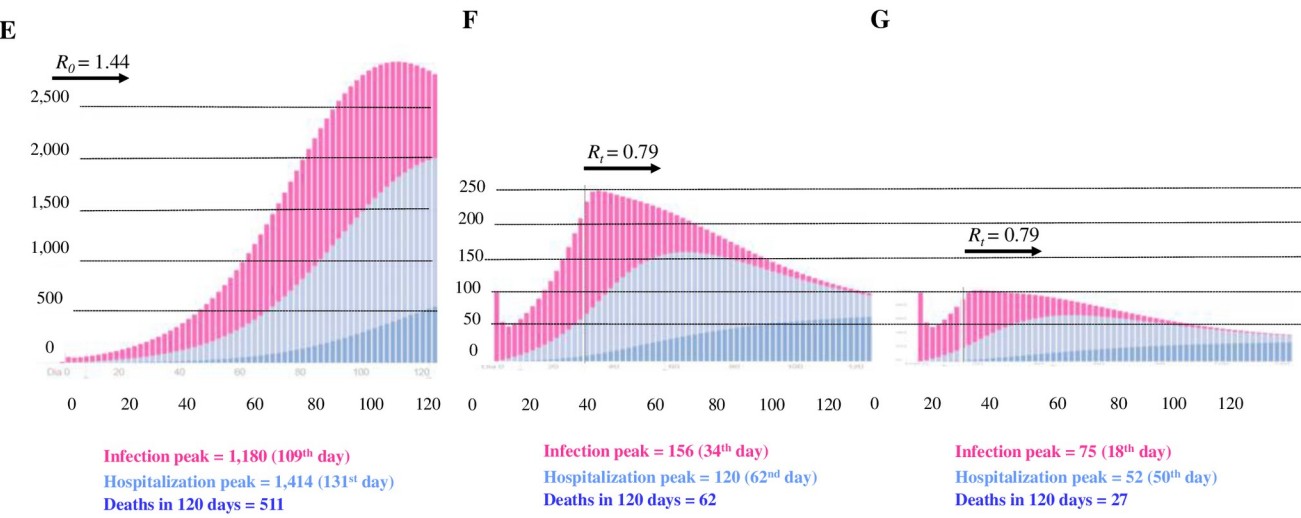

**A**

n° of COVID-19 cases

- Total COVID-19 cases
- Exponencial prediction
- Linear prediction

277
1091
424
120
23
35

**B**

$R_0 = 2.79$

Infection peak = 6,613 (40th day)
Hospitalization peak = 4,127 (62nd day)
Deaths in 120 days = 1,666

**C**

$R_t = 1.40$

Infection peak = 5,511 (34th day)
Hospitalization peak = 3,149 (60th day)
Deaths in 120 days = 1,316

**D**

$R_t = 1.40$

Infection peak = 1,314 (66th day)
Hospitalization peak = 1,582 (85th day)
Deaths in 120 days = 862

**E**

$R_0 = 1.44$

Infection peak = 1,180 (109th day)
Hospitalization peak = 1,414 (131st day)
Deaths in 120 days = 511

**F**

$R_t = 0.79$

Infection peak = 156 (34th day)
Hospitalization peak = 120 (62nd day)
Deaths in 120 days = 62

**G**

$R_t = 0.79$

Infection peak = 75 (18th day)
Hospitalization peak = 52 (50th day)
Deaths in 120 days = 27

**Fig 3. Predictions for COVID-19 cases, hospitalizations, and deaths in Ijuí, Brazil. A)** Difference between actual and predicted cases of COVID-19 using linear regression (green line) and exponential equation (blue line). **B)** Predictions without any new intervention after the 100th case ($R_0$ = 2.79). **C)** Predictions after implementing strategies to reduce the transmission rate by 50% 30 days after the 100th case ($R_t$ = 1.40). **D)** Predictions after implementing strategies to reduce the transmission rate by 50% 15 days after the 100th case ($R_t$ = 1.40). **E)** Predictions without any new intervention after the 100th case ($R_0$ = 1.44). **F)** Predictions after implementing strategies to reduce the transmission rate by 50% 30 days after the 100th case ($R_t$ = 0.79). **G)** Predictions after implementing strategies to reduce the transmission rate by 50% 15 days after the 100th case ($R_t$ = 0.79). All predictions were performed in the free epidemic simulator app from São Paulo University. Available at: https://ciis.fmrp.usp.br/covid19/epcalc/public/index.html.

commercial businesses were allowed to operate, but only at a ratio of one customer per employee, respecting the limits established in the occupational and operational protocols. Sports clubs were allowed to operate exclusively for the physical conditioning of the respective contracted professional athletes, observing the minimum distance of two meters between them. Physical contact or agglomerations were prohibited in all public settings [19]. Thus, the levels social distancing we measured in Ijuí may have been in accordance with these official recommendations, but were nevertheless insufficient to prevent a rapid increase in the number of COVID-19 cases. (For Ijuí urban geographic details, please see S1 Fig in S1 File, with geographic localization of COVID-19 cases in June and August).

Social distancing measures appear effective, mainly when implemented in conjunction with the isolation of people who test positive for COVID-19 and quarantining of anyone who has been in contact with them [31, 37]. Our data indicates that that preventive behavior among the population of Ijuí, related to the SDA recommendation and DPR, did not reach 70% participation in social distancing in any of the five waves of the survey. We mentioned the threshold of 70% since it was proposed early that maintaining social distancing at a maximum of 76% could prevent 90,000 COVID-19-related deaths and keep intensive care units in São Paulo from being overwhelmed [38]. As of November 23, 2020, São Paulo has registered more than 394,000 COVID-19 cases and 14,000 deaths [7], suggesting that maintaining and strengthening current social distancing measures, isolating COVID cases, and quarantining people who have been in contact with others who have tested positive, is absolutely vital to avoid serious stress to Brazil's healthcare system. Furthermore, it has been suggested that more restrictive recommendations can be more effective in reducing the number of infected subjects [1, 38–41], and it is necessary to apply such recommendations immediately and rigorously, especially to control the spread of COVID-19 in schools, since children and teenagers may have a disproportional contribution to an increase in the transmission rates [18]. However, as reported by WHO, many countries have reported an increase in "pandemic fatigue" among the population, characterized by lack of motivation to follow the recommended social distancing behaviors to protect themselves and others from the virus (WHO) [42]. On March 13, 2020 the United Stated issued a national proclamation that almost immediately resulted in a large number of people sheltering at home and reducing their daily movements, in line with the MSDI trends we observed in Ijuí in March. From early April to mid-April, the MSDI reached an upper limit followed by a plateau, indicating "social distancing inertia" in Ijuí. After that, a reduction in social distancing measures occurred even in the states that maintained the recommendation of mobility restriction, an example of "quarantine fatigue" [17].

Alongside this, considering data from Middle East respiratory syndrome (MERS) in Saudi Arabia in 2014 and SARS caused by SARS-CoV-1 in China in 2003, the psychological effects of a new pandemic tend to be more pronounced, widespread, and longer-lasting than the pure somatic effects of the infection, and the "epidemic of the fear" may be worse than the disease itself. It has been estimated anxiety about the possibility of infection ranges from 24% to 83% at the beginning of an epidemic, while the long-term epidemic period may trigger or

exacerbate stress-related mental disorders such as mood disorders, anxiety disorders, and post-traumatic stress disorder [18].

An intrinsic limitation of our study is the variability of self-perception about COVID-19 risk and what it means to socially distance. People see and act in different ways, depending on whether those things are perceived as psychologically relatable [43]. Thus, people respond to social distancing recommendations according to their empirical constructs. The construal level theory (CLT) of psychological distance has been recognized as a way to discuss the judgment and decision-making related to distance perception, which comprehends a mutual meaning of distance dimensions: temporal distance, social distance, spatial distance, and hypotheticality (i.e., distance from actuality) [44]. As a consequence, individual constructs about distance may influence evaluation, prediction, and behavior. In the same way, individuals may have different conceptions about risk, even if we consider risk directly as the chance of injury, damage, or loss [45]. As for the concept of temporal distance, Li and colleagues [46] suggested that people answering the survey could be influenced by the immediate pandemic-related context and details. Furthermore, because COVID-19 is primarily transmitted through close contact, people are more sensitive to implement social distancing with strangers and tend to believe that their behavior is in accordance with municipal social distancing regulations [46].

In this scenario, perceptions of risk play a key role in a process called "social amplification of risk." Social amplification of risk is triggered by the occurrence of an adverse event (whether major or minor) and reflects the fact that the adverse impacts of such an event sometimes extend far beyond the direct damages to victims and may result in massive indirect impacts [45]. Also, extensive media coverage of an event can contribute to heightened perceptions of risk and may have influenced the answers in our survey. Recent studies have shown that factors such as gender, race, political views, affiliations, emotional affect, and trust are strongly correlated with risk perception. Equally important is that these factors can influence the judgments of experts as well as laypeople [18, 45]. Our study is limited in its ability to investigate psychological influences during COVID-19 pandemic period, and further studies regarding the "feelings of subjects" about social distancing are recommended to better understand the phenomenon of "fatigue quarantine behavior" worldwide.

At the end of June 2020, Brazil reached more than 1.4 million COVID-19 cases [7, 35]. On June 30, RS reported 26,941 cases, or 344 cases per 100,000 inhabitants [8]. From June to July 2020, the hospitalization rate in RS decreased from 13% to 11%, while the proportion of COVID-19 deaths among hospitalized patients rose from 2.3 to 2.5% (currently 8.5 per 100,000 inhabitants). If Ijuí had the same rates of cases, hospitalization, and deaths as RS in June, it would have 287 cases, 31 hospitalizations, and seven deaths. In reality Ijuí had 283 cases, 27 hospitalizations, and one death. Ijuí ended the month of June with 80.8% of intensive care unit beds occupied and three COVID-19 patients [8, 19].

In our "optimistic" prediction (reducing transmission by 50%, $R = 0.79$, for 120 days after the 100[th] COVID-19 case) the infection peak was predicted to have 75 simultaneous COVID-19 cases and 52 hospitalizations. However, at its peak (August 6, 2020, ~60 days after the 100[th] COVID-19 case) Ijuí had 156 COVID-19 cases and eight hospitalized patients (the total number of hospitalized patients was 49, or 9.4% of COVID-19 cases). Our predictions also showed that 120 days after the 100[th] COVID-19 case, Ijuí would have a total of 27 deaths. Until August 6, 2020 Ijuí had only five deaths, allowing us to infer that the population at least partially followed the recommendations to avoid the transmission of the coronavirus and also reflects improvements in medical knowledge about COVID-19 and better ways of treating patients. In fact, 120 days after the 100[th] COVID-19 case, Ijuí reported a total of 25 deaths, closer to our prediction. The worst period of the COVID-19 pandemic in Ijuí (as of November 2020) was in

September and October. Cases and deaths have continued to accumulate with a sustained number of 100–183 cases per week for seven consecutive weeks) reaching 2,153 cases and 38 deaths in the middle of November. This sustained transmission of COVID-19 indicates that our model failed to predict the real behavior of the community over 120 days even if was able to correctly predict the number of COVID-19 deaths.

Up-to-date COVID-19 worldwide data preliminary indicates an increased risk for developing a severe form of COVID-19 in people with comorbidities, advanced age, or who are male. Pre-existing comorbidities were present in about half of patients with the severe form of disease; 30% had hypertension, 19% had diabetes, 8% had coronary heart disease, and 3% had a previous pulmonary condition such as chronic obstructive lung disease [47, 48]. COVID-19 patients with these comorbidities were also among those with highest mortality rates, with an adjusted OR of 7.42 (95% CI: 6.33–8.79) for people with hypertension, 9.03 (95% CI: 7.39–11.35) for people with diabetes, 12.83 (95% CI: 10.27–15.86) for people with coronary heart disease, and 7.79 (95% CI: 5.54–10.43) for chronic obstructive lung disease [49]. Furthermore, 31% of COVID-19 cases, 45% of hospitalizations, 53% of intensive care unit admissions, and 80% of deaths occurred among subjects aged ≥65 years [50]. Similarly, the case-fatality rates among individuals aged ≥80 years are approximately 20% [51]. The discrepancy between males and females is also noteworthy: Men account for 60% of hospital admissions and 70% of hospital deaths [1]. In the Lombardy Region of Italy, these figures are even more exaggerated: 82% of the patients admitted to intensive care units were older men [50]. This risk profile (i.e. male, older, and with existing cardiovascular disease) may be related to an impaired immune-metabolism stress response. Thus, these subjects cannot resolve virus-induced inflammatory bursts physiologically and are susceptible to exacerbated forms of inflammation [52], leading to a fatal "cytokine storm" [53, 54]. Thus, in future studies it is necessary to examine adherence to protective social distancing behaviors together with the presence of comorbidities [3, 55]. Greater engagement in preventive behavior among older subjects and people with chronic diseases in the community may reduce hospitalization and mortality rates.

## Conclusion

The insufficient engagement in social distancing behavior registered in this population-based study in Ijuí, Brazil may be related to the rapid increase of COVID-19 cases in this city. Our data predict an approaching outbreak of community spread of COVID-19, which could be avoided or attenuated if the levels of the social distancing among the population increase in the coming weeks.

## Supporting information

**S1 File.**
(PDF)

## Acknowledgments

We would like to thank all volunteers who participated in the population-based survey. We would also like to thank Professor Airam Sausen for her courses on COVID-19 prediction math.

## Author Contributions

**Conceptualization:** Thiago Gomes Heck, Rafael Z. Frantz, Mirna Stela Ludwig, Evelise Moraes Berlezi.

**Data curation:** Thiago Gomes Heck, Rafael Z. Frantz, Marilia Arndt Mesenburg, Lígia Beatriz Bento Franz.

**Formal analysis:** Thiago Gomes Heck, Marilia Arndt Mesenburg.

**Funding acquisition:** Thiago Gomes Heck, Evelise Moraes Berlezi.

**Investigation:** Thiago Gomes Heck, Matias Nunes Frizzo, Carlos Henrique Ramires François, Evelise Moraes Berlezi.

**Methodology:** Thiago Gomes Heck, Matias Nunes Frizzo, Evelise Moraes Berlezi.

**Project administration:** Thiago Gomes Heck.

**Resources:** Thiago Gomes Heck, Giovano Pereira Buratti.

**Software:** Thiago Gomes Heck, Rafael Z. Frantz.

**Supervision:** Thiago Gomes Heck.

**Validation:** Thiago Gomes Heck, Mirna Stela Ludwig, Evelise Moraes Berlezi.

**Visualization:** Thiago Gomes Heck.

**Writing – original draft:** Thiago Gomes Heck, Rafael Z. Frantz, Mirna Stela Ludwig.

**Writing – review & editing:** Thiago Gomes Heck, Rafael Z. Frantz, Matias Nunes Frizzo, Carlos Henrique Ramires François, Mirna Stela Ludwig, Marilia Arndt Mesenburg, Giovano Pereira Buratti, Lígia Beatriz Bento Franz, Evelise Moraes Berlezi.

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
