## [Decision Letter · Decision Letter 0]

19 Oct 2020

PONE-D-20-19438

Insufficient social distancing may be related to COVID-19 outbreak: the case of Ijuí city in Brazil

PLOS ONE

Dear Dr. Heck,

Thank you for submitting your manuscript to PLOS ONE. After careful consideration, we feel that it has merit but does not fully meet PLOS ONE’s publication criteria as it currently stands. Therefore, we invite you to submit a revised version of the manuscript that addresses the points raised during the review process.

We look forward to receiving your revised manuscript.

Kind regards,

Amir H. Pakpour, Ph.D.

Academic Editor

PLOS ONE

Journal Requirements:

4. In the Methods, please discuss whether and how the questionnaire was validated and/or pre-tested. If this did not occur, please provide the rationale for not doing so.

5.Thank you for stating the following in the Acknowledgments Section of your manuscript:

[This work was supported by the

371 Regional University of Northwestern Rio Grande do Sul State (UNIJUI), Federal University of Pelotas

372 (UFPEL), and Government of Rio Grande do Sul State, as well as by the Coordination for the Improvement

373 of Higher Education Personnel (CAPES).]

 [The author(s) received no specific funding for this work.]

6. We note that Supplementary Figure S1 in your submission contain [map/satellite] images which may be copyrighted. All PLOS content is published under the Creative Commons Attribution License (CC BY 4.0), which means that the manuscript, images, and Supporting Information files will be freely available online, and any third party is permitted to access, download, copy, distribute, and use these materials in any way, even commercially, with proper attribution. For these reasons, we cannot publish previously copyrighted maps or satellite images created using proprietary data, such as Google software (Google Maps, Street View, and Earth). For more information, see our copyright guidelines: http://journals.plos.org/plosone/s/licenses-and-copyright.

1.    You may seek permission from the original copyright holder of Supplementary Figure S1 to publish the content specifically under the CC BY 4.0 license. 

Reviewers' comments:

Reviewer's Responses to Questions

**Comments to the Author**

1. Is the manuscript technically sound, and do the data support the conclusions?

Reviewer #1: Yes

2. Has the statistical analysis been performed appropriately and rigorously? 

Reviewer #1: Yes

3. Have the authors made all data underlying the findings in their manuscript fully available?

Reviewer #1: Yes

4. Is the manuscript presented in an intelligible fashion and written in standard English?

Reviewer #1: No

5. Review Comments to the Author

Reviewer #1: Ref.: Insufficient social distancing may be related to COVID-19 outbreak: the case of Ijuí city (RS) in Brazil

Comments and suggestions:

The study seems to me to be well written, with the data well summarized and presenting and discussing a set of data can be interesting to understand the role of social distance in the management, mitigation, of the pandemic outbreak, particularly in the case of the city of Ijuí, a of the most populous in the state of Rio Grande do Sul, Brazil.

However, I would like to point out some specific and general aspects and comments, which I understand, could enrich and clarify the content of the study.

Specific comments:

1. In the introductory part of the manuscript, starting at line 54, it would be interesting for the authors to update the data by comparing some indicators of cases and deaths, especially with other Brazilian states. Perhaps, with the States of São Paulo, Rio de Janeiro and Ceará for which various indicators have always been the highest in Brazil. Perhaps, an illustrative table or figure can be used showing a relative comparative configuration of the varied incidence indicators (free choice by the authors);

2. It would also be interesting to describe and characterize Brazil, as a whole <showing 27="" a="" is="" it="" nation="" states="" that="" with="">3. It is also interesting to show how the indicators of social distance, control and measurement, varied weekly (choose the unit of analysis), or over weekends when compared to working days. I recommend, see Prof. Steven Taylor, Psychology of Pandemics, .... 2020);

4. From line 71, I believe that the description of the distance is insufficient. I recommend doing a brief review of the literature (there are several papers on this topic) about the role of social detachment, social isolation and other behavioral measures (the only ones that are effective while the vaccine or other collective immunization process is absent). The literature and the importance highlighted between lines 71 to 78 are quite limited in view of the large number of studies (papers) hitherto published on the role of isolation and social distance in controlling the spread of Covid-19;

5. Between lines 112 to 121, one could (if not inserted before) a table comparing the weekly variation or Regina comparison within the State of RS of the social distancing indicator (MSDI), as well as on its validity and reliability ( accuracy, reliability, as a behavioral indicator);

6. Somewhere in the description and characterization of the sample, population studied, it could be, in my best consideration, insert a table describing the population characteristics of those considered in the studies in some of its phases, especially responding to surveys, questionnaires and other sociodemographic information. Brazil and its 27 states are very unequal considering any indicator of inequality, for example, the GINI index or even HDI-regional or state, municipal;

General comments:

1. I think that the authors should consider some other variables that may be underlying the effects of social detachment, or social isolation, such as the personality variables of individuals and the social psychological variables that affect, overly, the attitudes and behaviors related to hygienic practices, cognitive assessment of risk factors, social distance and social isolation, as well as when to implement, if this occurs, mass, collective vaccination.

2. Spatial distance, temporal distance, social distance and probability distance (heuristic availability) could be considered in passing in the analysis of the results, especially in the discussion. There is an immense literature showing that these psychological distances affect the perceived risk and affect the cognitive assessment of the importance of the immediate introduction of distance and social isolation;

3. Data analysis is clearly described and summarized in figures that are very revealing;

4. Could the authors discuss the downward trend in the social distancing indicators that are becoming evident as the pandemic outbreak has lengthened? Was there a psychological accommodation, or a pandemic fatigue?

5. I understand that the role of social detachment should at least be discussed in the important light of psychological factors of personality and psychological social factors. Behaviors are the genuine tools we have to manage and mitigate the pandemic outbreak;

6. Statistical and epidemiological analyzes are appropriate and well described;

7. The work, in my view, constitutes an important study to understand the pandemic spread.

And my opinion,

October 14, 2020

José Aparecido Da Silva

Full Professor</showing>

6. PLOS authors have the option to publish the peer review history of their article (what does this mean?). If published, this will include your full peer review and any attached files.

Reviewer #1: No

---

## [Author Response · Author response to Decision Letter 0]

24 Nov 2020

To Prof. Amir H. Pakpour, Ph.D.

Academic Editor

PLOS ONE

Ijuí, November 24th, 2020.

Dear Prof. Pakpour,

While we are grateful for the opportunity of presenting a revised version of our manuscript entitled “Insufficient social distancing may be related to COVID-19 outbreak: the case of Ijuí city in Brazil”, ID PONE-D-20-19438. We hereby would like to present our responses to the reviewers’ comments. All the alterations to the text are now red-marked to facilitate reviewers’ analyses. We are also uploading a clean version of the manuscript. 

JOURNAL REQUIREMENTS:

Q1. Please ensure that your manuscript meets PLOS ONE's style requirements, including those for file naming. The PLOS ONE style templates can be found at 

Answer 1. Thank you for the comment and sorry for the mistakes in the first version of the manuscript. We revised the manuscript in terms of PLOS ONE’s style requirements. The revised version has now the correct formatting of the head titles levels. Also, we changed “Figure 1” by “Fig 1” as recommended, as well as we corrected the information about authors’ affiliations.

Q2. We suggest you thoroughly copyedit your manuscript for language usage, spelling, and grammar. If you do not know anyone who can help you do this, you may wish to consider employing a professional scientific editing service. 

Whilst you may use any professional scientific editing service of your choice, PLOS has partnered with both American Journal Experts (AJE) and Editage to provide discounted services to PLOS authors. Both organizations have experience helping authors meet PLOS guidelines and can provide language editing, translation, manuscript formatting, and figure formatting to ensure your manuscript meets our submission guidelines. To take advantage of our partnership with AJE, visit the AJE website (http://learn.aje.com/plos/) for a 15% discount off AJE services. To take advantage of our partnership with Editage, visit the Editage website (www.editage.com) and enter referral code PLOSEDIT for a 15% discount off Editage services. If the PLOS editorial team finds any language issues in text that either AJE or Editage has edited, the service provider will re-edit the text for free. Upon resubmission, please provide the following:

The name of the colleague or the details of the professional service that edited your manuscript A copy of your manuscript showing your changes by either highlighting them or using track changes (uploaded as a *supporting information* file). A clean copy of the edited manuscript (uploaded as the new *manuscript* file)

Answer 2. Thank you for the comment and sorry for some English mistakes in the first version of the manuscript. We sent the manuscript to Cambridge Proofreading service (https://proofreading.org/). Now, we have a revised version of the manuscript, that we believe reached language quality after it was revised by English native speaker and professional proof-read service in the field. Thus, we modified the title from “Insufficient social distancing may be related to COVID-19 outbreak: the case of Ijuí city in Brazil” to “Insufficient social distancing may contribute to COVID-19 outbreak: the case of Ijuí city in Brazil”. Also, the marked English revision was attached to the online submission system. The certificate of English revision is attached below: 

Q3. Please include additional information regarding the survey or questionnaire used in the study and ensure that you have provided sufficient details that others could replicate the analyses. For instance, if you developed a questionnaire as part of this study and it is not under a copyright more restrictive than CC-BY, please include a copy, in both the original language and English, as Supporting Information.

Answer 3. Thank you for the recommendation. We included a description of the questions related to social distancing behavior to provide a better comprehension for readers and to provide an opportunity for replication of the study in other cities. Also, we included in the supplementary material the questionary in English and Portuguese. Thank you for the suggestion. Also, we provide a link for data availability as now mentioned at the end of manuscript. These information were described as follows:

Page 6, line 163: 

“Additionally, at each study wave participants answered short questionnaires, including sociodemographic information (sex, age, medical history, schooling, and race), COVID-19-related symptoms, use of health services, compliance with social distancing measures, and use of face masks. The questions on social distancing were as follows: 1) “To what extent are you managing to follow the social distancing guidance from the health authorities, i.e., staying at home and avoiding contact with others?” This was scored on a five-point scale, with the following alternatives read aloud to the respondent: “very little,” “little,” “some,” “quite,” and “practically isolated from everyone;” 2) “What have your routine activities been?” The alternatives were: “staying home all the time,” “only leaving home only for essentials, such as groceries,” “leaving home from time to time to run errands and stretch legs,” “going out every day for regular activities,” and “out of the house all day, every day, either for work or for other regular activities.” This questionnaire passed an internal validation before it was applied in this study. After this, the questionnaire was applied in 133 cities covering all regions of Brazil [6,24–26]. The dataset used to produce the analyses presented in this study is freely available at http://www.rs.epicovid19brasil.org/banco-de-dados/ and from the corresponding author upon request. The questionnaire is available in the supplementary material.” 

Q4. In the Methods, please discuss whether and how the questionnaire was validated and/or pre-tested. If this did not occur, please provide the rationale for not doing so.

Answer 4. The questionary was elaborated by experts from Epidemiology research group from the Federal University of Pelotas, coordinated by the researcher Prof. Pedro Hallal. The questionary passed by an internal validation before initiates the field steps. The field work in Ijuí was carried out by Instituto de Pesquisas de Opinião, a contract research organization, with the aid of UNIJUI university. The interviewers were selected among students of health graduate and undergraduate programs from UNIJUI. All were trained in administering the questionnaire. The questionnaire was included in the supplementary material. After the first rounds of study in the RS, this questionary was used in 133 Brazilian cities to evaluate the COVID-19 scenario in the country. Due to the emergency to know about the community transmission rates, the survey was applied in the population just after an internal validation of the questionary.

Page 6, line 163: 

“Additionally, at each study wave participants answered short questionnaires, including sociodemographic information (sex, age, medical history, schooling, and race), COVID-19-related symptoms, use of health services, compliance with social distancing measures, and use of face masks. The questions on social distancing were as follows: 1) “To what extent are you managing to follow the social distancing guidance from the health authorities, i.e., staying at home and avoiding contact with others?” This was scored on a five-point scale, with the following alternatives read aloud to the respondent: “very little,” “little,” “some,” “quite,” and “practically isolated from everyone;” 2) “What have your routine activities been?” The alternatives were: “staying home all the time,” “only leaving home only for essentials, such as groceries,” “leaving home from time to time to run errands and stretch legs,” “going out every day for regular activities,” and “out of the house all day, every day, either for work or for other regular activities.” This questionnaire passed an internal validation before it was applied in this study. After this, the questionnaire was applied in 133 cities covering all regions of Brazil [6,24–26]. The dataset used to produce the analyses presented in this study is freely available at http://www.rs.epicovid19brasil.org/banco-de-dados/ and from the corresponding author upon request. The questionnaire is available in the supplementary material.” 

Q5. Thank you for stating the following in the Acknowledgments Section of your manuscript:

[This work was supported by the 371 Regional University of Northwestern Rio Grande do Sul State (UNIJUI), Federal University of Pelotas 372 (UFPEL), and Government of Rio Grande do Sul State, as well as by the Coordination for the Improvement 373 of Higher Education Personnel (CAPES).]

We note that you have provided funding information that is not currently declared in your Funding Statement. However, funding information should not appear in the Acknowledgments section or other areas of your manuscript. We will only publish funding information present in the Funding Statement section of the online submission form. Please remove any funding-related text from the manuscript and let us know how you would like to update your Funding Statement. Currently, your Funding Statement reads as follows:

[The author(s) received no specific funding for this work.]

Answer 5. Thank you for the recommendation. We deleted the funding information in the Acknowledgments Section. Now, we have this text:

Page 17 line 460

Acknowledgments

We would like to thank all volunteers who participated in the population-based survey. We would also like to thank Professor Airam Sausen for her courses on COVID-19 prediction math.

Q6. We note that Supplementary Figure S1 in your submission contain [map/satellite] images which may be copyrighted. All PLOS content is published under the Creative Commons Attribution License (CC BY 4.0), which means that the manuscript, images, and Supporting Information files will be freely available online, and any third party is permitted to access, download, copy, distribute, and use these materials in any way, even commercially, with proper attribution. For these reasons, we cannot publish previously copyrighted maps or satellite images created using proprietary data, such as Google software (Google Maps, Street View, and Earth). For more information, see our copyright guidelines: http://journals.plos.org/plosone/s/licenses-and-copyright.

1. You may seek permission from the original copyright holder of Supplementary Figure S1 to publish the content specifically under the CC BY 4.0 license. 

Please upload the completed Content Permission Form or other proof of granted permissions as an "Other" file with your submission. In the figure caption of the copyrighted figure, please include the following text: “Reprinted from [ref] under a CC BY license, with permission from [name of publisher], original copyright [original copyright year].”

Answer 6 Thank you for the recommendation. We modified the figure using the recommended link https://viewer.nationalmap.gov/advanced-viewer. The new figure is showed below:

Q7. Please include captions for your Supporting Information files at the end of your manuscript, and update any in-text citations to match accordingly. Please see our Supporting Information guidelines for more information: http://journals.plos.org/plosone/s/supporting-information.

Answer 7. Thank you for the correction. We revised all figures and tables citations, legends and information according to PLOS One rules.

ANSWERS TO THE REVIEWERS

Review Comments to the Author

Reviewer #1: 

Comments and suggestions:

The study seems to me to be well written, with the data well summarized and presenting and discussing a set of data can be interesting to understand the role of social distance in the management, mitigation, of the pandemic outbreak, particularly in the case of the city of Ijuí, a of the most populous in the state of Rio Grande do Sul, Brazil. However, I would like to point out some specific and general aspects and comments, which I understand, could enrich and clarify the content of the study.

Specific comments:

Q8. In the introductory part of the manuscript, starting at line 54, it would be interesting for the authors to update the data by comparing some indicators of cases and deaths, especially with other Brazilian states. Perhaps, with the States of São Paulo, Rio de Janeiro and Ceará for which various indicators have always been the highest in Brazil. Perhaps, an illustrative table or figure can be used showing a relative comparative configuration of the varied incidence indicators (free choice by the authors);

Answer 8. Thank you for your suggestion. In fact, since our first version of the manuscript (august) for nowadays, we have more clear data regarding COVID-19 transmission in different States and cities in Brazil. We included a short description in the introduction to provide an overview of Brazil situation. 

Page 3 line 54

“The first case of COVID-19 in Brazil was reported on February 27, 2020 in the city of São Paulo. Based on published events, eight of the 27 federated units of Brazil present cumulative mortality rates above 10 per 100,000 inhabitants: four in the north, two in the northeast, and two in the southeast region (including Rio de Janeiro and São Paulo) [6]. Until November 2020, Brazil officially recorded 5,468,270 cases of COVID-19 (2,602 per 100,000 inhabitants) and 158,456 COVID-19 deaths (75 per 100,000 inhabitants). The five federative units with the highest mortality counts are São Paulo (39,007 deaths), Rio de Janeiro (20,376 deaths), Ceará (9,325 deaths), Minas Gerais (8,872 deaths), and Pernambuco (8,587 deaths). The highest cumulative mortality rates above 10 per 100,000 inhabitants are found in Ceará, with 102 deaths per 100,000 inhabitants[7]. In the state of Rio Grande do Sul (RS), the southernmost state in Brazil with 11.3 million people, the first case of COVID-19 was diagnosed on February 29, 2020. As of August 6, 2020, 76,563 confirmed cases (673 per 100,000 inhabitants) and 2,163 deaths (19 per 100,000 inhabitants, 2.8% of confirmed cases) have been reported [8,9]. As of November 2020, RS recorded 240,694 COVID-19 cases (2,116 per 100,000 inhabitants) and 5,699 deaths (50 per 100,000 inhabitants)[8].” 

Q9. It would also be interesting to describe and characterize Brazil, as a whole 3. It is also interesting to show how the indicators of social distance, control and measurement, varied weekly (choose the unit of analysis), or over weekends when compared to working days. I recommend, see Prof. Steven Taylor, Psychology of Pandemics, .... 2020);

Answer 9. Thank you for your recommendation and suggestion. We modified different parts of our manuscript to attend your good suggestions. Also, thank you so much by the recommendation of the Taylor’s book. We included information from that book and other related references in different parts of our manuscript. This is a very interesting material for more than support our study. Also, we described in details in the results, the descriptive data about social distancing including mean, standard deviation, minimum and maximum and we compared data from working days vs weekend. We found a difference between these days. Thank you for your suggestion. We included a short description about Brazil differences in the discussion section as follows:

Introduction, Page 3 line 67

“Although a significant investment has been made worldwide to provide antiviral prophylaxis for COVID-19, to test different drugs for prevention or treatment COVID-19 cases, and to develop vaccines [10], current recommendations to reduce the spread of COVID-19 include physical distancing [11], quarantining, and large-scale lockdowns of entire populations [12][13]. Evidence indicates that the implementation of social distancing can suppress COVID-19 transmission rates to prevent the disease from overwhelming the healthcare system. In an analysis of 49 countries, Atalan [14] showed that the COVID-19 pandemic can be suppressed by lockdown measures. In another study including data from 131 countries [15], a decrease in the transmission rate of COVID-19 was observed within 1-3 weeks following the introduction of school closures, workplace closures, public events bans, stay-at-home orders, and limits on internal movement. However, the reduction of transmission ranged from 3% to 24% approximately one month following the introduction of the recommendations, and the effect was only statistically significant for public events bans [15]. Similarly, in New Zealand (a country of 4.886 million inhabitants), the estimated COVID-19 case infection rate decreased from 8.5 to 3.2 per one million people after the implementation of a nationwide the lockdown, resulting in a low relative burden of disease [16]; until now New Zealand has accumulated only 1,973 COVID-19 cases and 25 deaths. Although social distancing and lockdown measures appear to be successful, there is “social fatigue” associated with following these recommendations, leading many societies to return to a usual lifes, increasing COVID-19 transmission [17,18].”

Results, Page 9 line 225

 “…For a fair comparison, percentages pointed above the open circles in Figure 1A correspond to MSDI on weekends, whereas the closed circles correspond to MSDI on working days. We observed an increase in distancing on the weekends, indicating that more people tended to stay home on weekends vs. weekdays. Specifically, on weekends MDSI was ~12% higher than on working days (44.87 ± 9.70 [95% CI: 41.99 - 47.75] vs. 36.07 ± 7.45 [95% CI: 34.66 - 37.48], p < 0.0001, Student’s t-test). However, a decrease in overall MSDI (weekends and weekdays) was registered between March 22, 2020 (70.2%) and June 21, 2020 (48.2%). From a macro point of view, it is evident that the population of Ijuí is loosening social distancing and increasing social interaction up to the end of June.” 

Discussion, page 13 line 359

“Social distancing measures appear effective, mainly when implemented in conjunction with the isolation of people who test positive for COVID-19 and quarantining of anyone who has been in contact with them [31,37]. Our data indicates that that preventive behavior among the population of Ijuí, related to the SDA recommendation and DPR, did not reach 70% participation in social distancing in any of the five waves of the survey. We mentioned the threshold of 70% since it was proposed early that maintaining social distancing at a maximum of 76% could prevent 90,000 COVID-19-related deaths and keep intensive care units in São Paulo from being overwhelmed [38]. As of November 23, 2020, São Paulo has registered more than 394,000 COVID-19 cases and 14,000 deaths [7], suggesting that maintaining and strengthening current social distancing measures, isolating COVID cases, and quarantining people who have been in contact with others who have tested positive, is absolutely vital to avoid serious stress to Brazil’s healthcare system. Furthermore, it has been suggested that more restrictive recommendations can be more effective in reducing the number of infected subjects [1,38–41], and it is necessary to apply such recommendations immediately and rigorously, especially to control the spread of COVID-19 in schools, since children and teenagers may have a disproportional contribution to an increase in the transmission rates [18]. However, as reported by WHO, many countries have reported an increase in “pandemic fatigue” among the population, characterized by lack of motivation to follow the recommended social distancing behaviors to protect themselves and others from the virus (WHO) [42]. On March 13, 2020 the United Stated issued a national proclamation that almost immediately resulted in a large number of people sheltering at home and reducing their daily movements, in line with the MSDI trends we observed in Ijuí in March. From early April to mid-April, the MSDI reached an upper limit followed by a plateau, indicating “social distancing inertia” in Ijuí. After that, a reduction in social distancing measures occurred even in the states that maintained the recommendation of mobility restriction, an example of “quarantine fatigue” [17].”

Q10. From line 71, I believe that the description of the distance is insufficient. I recommend doing a brief review of the literature (there are several papers on this topic) about the role of social detachment, social isolation and other behavioral measures (the only ones that are effective while the vaccine or other collective immunization process is absent). The literature and the importance highlighted between lines 71 to 78 are quite limited in view of the large number of studies (papers) hitherto published on the role of isolation and social distance in controlling the spread of Covid-19;

Answer 10. Thank you for the recommendation. We improved the introduction and the discussion in this version of the manuscript, as follows:

Introduction, Page 3 line 67

“Although a significant investment has been made worldwide to provide antiviral prophylaxis for COVID-19, to test different drugs for prevention or treatment COVID-19 cases, and to develop vaccines [10], current recommendations to reduce the spread of COVID-19 include physical distancing [11], quarantining, and large-scale lockdowns of entire populations [12][13]. Evidence indicates that the implementation of social distancing can suppress COVID-19 transmission rates to prevent the disease from overwhelming the healthcare system. In an analysis of 49 countries, Atalan [14] showed that the COVID-19 pandemic can be suppressed by lockdown measures. In another study including data from 131 countries [15], a decrease in the transmission rate of COVID-19 was observed within 1-3 weeks following the introduction of school closures, workplace closures, public events bans, stay-at-home orders, and limits on internal movement. However, the reduction of transmission ranged from 3% to 24% approximately one month following the introduction of the recommendations, and the effect was only statistically significant for public events bans [15]. Similarly, in New Zealand (a country of 4.886 million inhabitants), the estimated COVID-19 case infection rate decreased from 8.5 to 3.2 per one million people after the implementation of a nationwide the lockdown, resulting in a low relative burden of disease [16]; until now New Zealand has accumulated only 1,973 COVID-19 cases and 25 deaths. Although social distancing and lockdown measures appear to be successful, there is “social fatigue” associated with following these recommendations, leading many societies to return to a usual lifes, increasing COVID-19 transmission [17,18].”

Discussion, Page 11 Line 298

“Although there are currently many studies about social distancing behavior and COVID-19 cases in Brazil, it is imperative to study local data separately. Brazil is a vast country comprised of 26 federative states and the Federal District, and there are many cultural, economic, educational, and geographic differences between states and between different cities in the same state. Given these differences, trends in the number of cases and deaths differs between states and cities, and municipalities have the autonomy to determine which measures to adopt in order to best mitigate COVID-19 according to their respective scenarios [33]. All of Brazil’s states implemented distancing measures, mostly after March 15, 2020. Partial economic lockdown was implemented before the tenth confirmed case of COVID-19 by 18 (67%) states and before the first death from COVID-19 by 24 (89%) of the states [33]. In April 2020, of nine major cities in Rio Grande do Sul, the biggest cities (Porto Alegre and Santa Maria) exhibited the highest degree of social distancing, while Ijuí had a less favorable pattern, with a strikingly higher percentage reported being “out of the house all day” [25].”

Discussion, Page 11 Line 369

“Furthermore, it has been suggested that more restrictive recommendations can be more effective in reducing the number of infected subjects [1,38–41], and it is necessary to apply such recommendations immediately and rigorously, especially to control the spread of COVID-19 in schools, since children and teenagers may have a disproportional contribution to an increase in the transmission rates [18]. However, as reported by WHO, many countries have reported an increase in “pandemic fatigue” among the population, characterized by lack of motivation to follow the recommended social distancing behaviors to protect themselves and others from the virus (WHO) [42]. On March 13, 2020 the United Stated issued a national proclamation that almost immediately resulted in a large number of people sheltering at home and reducing their daily movements, in line with the MSDI trends we observed in Ijuí in March. From early April to mid-April, the MSDI reached an upper limit followed by a plateau, indicating “social distancing inertia” in Ijuí. After that, a reduction in social distancing measures occurred even in the states that maintained the recommendation of mobility restriction, an example of “quarantine fatigue” [17]. 

Alongside this, considering data from Middle East respiratory syndrome (MERS) in Saudi Arabia in 2014 and SARS caused by SARS-CoV-1 in China in 2003, the psychological effects of a new pandemic tend to be more pronounced, widespread, and longer-lasting than the pure somatic effects of the infection, and the “epidemic of the fear” may be worse than the disease itself. It has been estimated anxiety about the possibility of infection ranges from 24% to 83% at the beginning of an epidemic, while the long-term epidemic period may trigger or exacerbate stress-related mental disorders such as mood disorders, anxiety disorders, and post-traumatic stress disorder [18]. 

An intrinsic limitation of our study is the variability of self-perception about COVID-19 risk and what it means to socially distance. People see and act in different ways, depending on whether those things are perceived as psychologically relatable [43]. Thus, people respond to social distancing recommendations according to their empirical constructs. The construal level theory (CLT) of psychological distance has been recognized as a way to discuss the judgment and decision-making related to distance perception, which comprehends a mutual meaning of distance dimensions: temporal distance, social distance, spatial distance, and hypotheticality (i.e., distance from actuality) [44]. As a consequence, individual constructs about distance may influence evaluation, prediction, and behavior. In the same way, individuals may have different conceptions about risk, even if we consider risk directly as the chance of injury, damage, or loss [45]. As for the concept of temporal distance, Li and colleagues [46] suggested that people answering the survey could be influenced by the immediate pandemic-related context and details. Furthermore, because COVID-19 is primarily transmitted through close contact, people are more sensitive to implement social distancing with strangers and tend to believe that their behavior is in accordance with municipal social distancing regulations [46]. 

In this scenario, perceptions of risk play a key role in a process called “social amplification of risk.” Social amplification of risk is triggered by the occurrence of an adverse event (whether major or minor) and reflects the fact that the adverse impacts of such an event sometimes extend far beyond the direct damages to victims and may result in massive indirect impacts [45]. Also, extensive media coverage of an event can contribute to heightened perceptions of risk and may have influenced the answers in our survey. Recent studies have shown that factors such as gender, race, political views, affiliations, emotional affect, and trust are strongly correlated with risk perception. Equally important is that these factors can influence the judgments of experts as well as laypeople [18,45]. Our study is limited in its ability to investigate psychological influences during COVID-19 pandemic period, and further studies regarding the “feelings of subjects” about social distancing are recommended to better understand the phenomenon of “fatigue quarantine behavior” worldwide.” 

Q11. Between lines 112 to 121, one could (if not inserted before) a table comparing the weekly variation or Regina comparison within the State of RS of the social distancing indicator (MSDI), as well as on its validity and reliability ( accuracy, reliability, as a behavioral indicator);

Answer 11. Thank you for the recommendation. We discussed our data in connection of the study of Oliveira and colleagues that provided interesting data with higher sample size than ours study. This study used a mobile phone isolation indexes in the biggest population states in Brazil, and described the validity of MSDI as a behavior indicator and its relation with covid19 transmission rates. The new paragraph is showed below:

Discussion, Page 12, line 316

“On March 19, 2020 the city administration issued its most restrictive decree in the analyzed period, The decree restricted public transport, closed stores, suspended classes at schools and universities, and established special protocols for restaurants and other services for one week [19]. Accordingly, on March 22, 2020 the MSDI reached its highest value: 70.2%. Other subsequent decrees were issued in Ijuí, but with more relaxed restrictions. In parallel, the Federal Government of Brazil has been minimizing this pandemic and, in most cases, encouraging people to keep their regular routines [34]. Thus, we observe a continuous reduction of the social distancing, to the point that at the end of June, MSDI fell to the same levels seen at the beginning of the analyzed period, when there were no cases of COVID-19 in Ijuí or even in Brazil as a whole [7,35]. Our results were similar to those reported by Oliveira and colleagues [36], presenting a mean isolation index from February 1, 2020 to April 10, 2020 of 40.2%, ranging from 18.5% to 69.4%. Specifically, this study analyzed mean isolation index in the states of São Paulo (13.5% to 67.9%) and Rio de Janeiro (16.6% to 69.4%) and found that social isolation indexes of 46.7% have the highest accuracy (93.9%) to predict R(t) <1 [36], which means that the epidemic is slowing. These data reinforce the validity and reliability of MSDI as a behavioral indicator of social distancing.”

Q12. Somewhere in the description and characterization of the sample, population studied, it could be, in my best consideration, insert a table describing the population characteristics of those considered in the studies in some of its phases, especially responding to surveys, questionnaires and other sociodemographic information. Brazil and its 27 states are very unequal considering any indicator of inequality, for example, the GINI index or even HDI-regional or state, municipal;

Answer 12. Thank you for the recommendation. We included more details about Ijuí HDI score in the methods. Also, we moved to the manuscript the results from tables 1 and 2, which was presented as supplementary material in the first version of the manuscript. These tables are about sociodemographic information and other characteristics of the sample involved in the survey as follows: 

Page 4 Line 106 

“Ijuí (28°23'16 S and 53°54'53" W) is the most populous city in the northwest region of Rio Grande do Sul. With 83,475 residents, it is considered a city of students (“university city”) and a center of hospital and university resources. Furthermore, it is the largest and most important population center in the region, with a population rounding 150,000 people. Ijuí has a high Human Development Index (HDI) score of 0.781, above the overall HDI of Brazil (0.761). Ijuí has a high score for all three parameters measured for HDI calculation: education (HDI-E = 0.707), with 98.9% of children aged 6-14 in school; longevity (HDI-L = 0.858), with an average life expectancy of 76.48 years; and per capita income (HDI-R = 0.786), with R$ 38,341.14 (approximately $7,119.33 per capita/year [23].”

Page 9 Line 245 

“In each round of the population-based survey, >400 adults were surveyed and tested for the presence of SARS-CoV-2 antibodies, for a total of 2,222 study participants. Characteristics of the study population are described in Tables 1 and 2. The majority of the subjects interviewed were women (~60%) and white (~80%), with a roughly equal distribution across age and education categories (Table 1). Between 32% and 39% of participants reported having hypertension and ~13% and ~10% had diabetes and asthma, respectively.” 

Table 1. Sociodemographic characteristics of Ijuí subjects by date of population-based survey 

Table 2. Comorbidities characteristics of Ijuí subjects by date of population-based survey

General comments:

Q13. I think that the authors should consider some other variables that may be underlying the effects of social detachment, or social isolation, such as the personality variables of individuals and the social psychological variables that affect, overly, the attitudes and behaviors related to hygienic practices, cognitive assessment of risk factors, social distance and social isolation, as well as when to implement, if this occurs, mass, collective vaccination.

Answer 13. We included in the manuscript a brief discussion about psychological influence in the meaning of the understanding about social distancing and risk. Since we considered that our studies had limitations in terms of deeper investigation of psychological variables, we inserted in the manuscript a recommendation to this type of study as follows:

Page 14 Line 388 

“An intrinsic limitation of our study is the variability of self-perception about COVID-19 risk and what it means to socially distance. People see and act in different ways, depending on whether those things are perceived as psychologically relatable [43]. Thus, people respond to social distancing recommendations according to their empirical constructs. The construal level theory (CLT) of psychological distance has been recognized as a way to discuss the judgment and decision-making related to distance perception, which comprehends a mutual meaning of distance dimensions: temporal distance, social distance, spatial distance, and hypotheticality (i.e., distance from actuality) [44]. As a consequence, individual constructs about distance may influence evaluation, prediction, and behavior. In the same way, individuals may have different conceptions about risk, even if we consider risk directly as the chance of injury, damage, or loss [45]. As for the concept of temporal distance, Li and colleagues [46] suggested that people answering the survey could be influenced by the immediate pandemic-related context and details. Furthermore, because COVID-19 is primarily transmitted through close contact, people are more sensitive to implement social distancing with strangers and tend to believe that their behavior is in accordance with municipal social distancing regulations [46]. 

In this scenario, perceptions of risk play a key role in a process called “social amplification of risk.” Social amplification of risk is triggered by the occurrence of an adverse event (whether major or minor) and reflects the fact that the adverse impacts of such an event sometimes extend far beyond the direct damages to victims and may result in massive indirect impacts [45]. Also, extensive media coverage of an event can contribute to heightened perceptions of risk and may have influenced the answers in our survey. Recent studies have shown that factors such as gender, race, political views, affiliations, emotional affect, and trust are strongly correlated with risk perception. Equally important is that these factors can influence the judgments of experts as well as laypeople [18,45]. Our study is limited in its ability to investigate psychological influences during COVID-19 pandemic period, and further studies regarding the “feelings of subjects” about social distancing are recommended to better understand the phenomenon of “fatigue quarantine behavior” worldwide.” 

Q14. Spatial distance, temporal distance, social distance and probability distance (heuristic availability) could be considered in passing in the analysis of the results, especially in the discussion. There is an immense literature showing that these psychological distances affect the perceived risk and affect the cognitive assessment of the importance of the immediate introduction of distance and social isolation;

Answer 14. Thank you for the recommendation. We included in the discussion section two paragraphs discussing concepts about distance and risk perception applying in the context of pandemic period as follows:

Page 14 Line 388 

“An intrinsic limitation of our study is the variability of self-perception about COVID-19 risk and what it means to socially distance. People see and act in different ways, depending on whether those things are perceived as psychologically relatable [43]. Thus, people respond to social distancing recommendations according to their empirical constructs. The construal level theory (CLT) of psychological distance has been recognized as a way to discuss the judgment and decision-making related to distance perception, which comprehends a mutual meaning of distance dimensions: temporal distance, social distance, spatial distance, and hypotheticality (i.e., distance from actuality) [44]. As a consequence, individual constructs about distance may influence evaluation, prediction, and behavior. In the same way, individuals may have different conceptions about risk, even if we consider risk directly as the chance of injury, damage, or loss [45]. As for the concept of temporal distance, Li and colleagues [46] suggested that people answering the survey could be influenced by the immediate pandemic-related context and details. Furthermore, because COVID-19 is primarily transmitted through close contact, people are more sensitive to implement social distancing with strangers and tend to believe that their behavior is in accordance with municipal social distancing regulations [46]. 

In this scenario, perceptions of risk play a key role in a process called “social amplification of risk.” Social amplification of risk is triggered by the occurrence of an adverse event (whether major or minor) and reflects the fact that the adverse impacts of such an event sometimes extend far beyond the direct damages to victims and may result in massive indirect impacts [45]. Also, extensive media coverage of an event can contribute to heightened perceptions of risk and may have influenced the answers in our survey. Recent studies have shown that factors such as gender, race, political views, affiliations, emotional affect, and trust are strongly correlated with risk perception. Equally important is that these factors can influence the judgments of experts as well as laypeople [18,45]. Our study is limited in its ability to investigate psychological influences during COVID-19 pandemic period, and further studies regarding the “feelings of subjects” about social distancing are recommended to better understand the phenomenon of “fatigue quarantine behavior” worldwide.”

Q15. Data analysis is clearly described and summarized in figures that are very revealing;

Answer 15. Thank very much for this comment. We hope that it can be useful for readers.

Q16. Could the authors discuss the downward trend in the social distancing indicators that are becoming evident as the pandemic outbreak has lengthened? Was there a psychological accommodation, or a pandemic fatigue?

Answer 16. Thank very much for this recommendation. We included the concepts and evidence about social distancing inertia and quarantine fatigue in the discussion as follows. 

Page 13 Line 359 

“Social distancing measures appear effective, mainly when implemented in conjunction with the isolation of people who test positive for COVID-19 and quarantining of anyone who has been in contact with them [31,37]. Our data indicates that that preventive behavior among the population of Ijuí, related to the SDA recommendation and DPR, did not reach 70% participation in social distancing in any of the five waves of the survey. We mentioned the threshold of 70% since it was proposed early that maintaining social distancing at a maximum of 76% could prevent 90,000 COVID-19-related deaths and keep intensive care units in São Paulo from being overwhelmed [38]. As of November 23, 2020, São Paulo has registered more than 394,000 COVID-19 cases and 14,000 deaths [7], suggesting that maintaining and strengthening current social distancing measures, isolating COVID cases, and quarantining people who have been in contact with others who have tested positive, is absolutely vital to avoid serious stress to Brazil’s healthcare system. Furthermore, it has been suggested that more restrictive recommendations can be more effective in reducing the number of infected subjects [1,38–41], and it is necessary to apply such recommendations immediately and rigorously, especially to control the spread of COVID-19 in schools, since children and teenagers may have a disproportional contribution to an increase in the transmission rates [18]. However, as reported by WHO, many countries have reported an increase in “pandemic fatigue” among the population, characterized by lack of motivation to follow the recommended social distancing behaviors to protect themselves and others from the virus (WHO) [42]. On March 13, 2020 the United Stated issued a national proclamation that almost immediately resulted in a large number of people sheltering at home and reducing their daily movements, in line with the MSDI trends we observed in Ijuí in March. From early April to mid-April, the MSDI reached an upper limit followed by a plateau, indicating “social distancing inertia” in Ijuí. After that, a reduction in social distancing measures occurred even in the states that maintained the recommendation of mobility restriction, an example of “quarantine fatigue” [17]. 

Alongside this, considering data from Middle East respiratory syndrome (MERS) in Saudi Arabia in 2014 and SARS caused by SARS-CoV-1 in China in 2003, the psychological effects of a new pandemic tend to be more pronounced, widespread, and longer-lasting than the pure somatic effects of the infection, and the “epidemic of the fear” may be worse than the disease itself. It has been estimated anxiety about the possibility of infection ranges from 24% to 83% at the beginning of an epidemic, while the long-term epidemic period may trigger or exacerbate stress-related mental disorders such as mood disorders, anxiety disorders, and post-traumatic stress disorder [18].”

Q17. I understand that the role of social detachment should at least be discussed in the important light of psychological factors of personality and psychological social factors. Behaviors are the genuine tools we have to manage and mitigate the pandemic outbreak;

Answer 17. Thank you for the recommendation. We included in the end of the discussion about the quarantine fatigue some psychological effects that may be a result of a pandemic period.

Page 13 Line 359 

“Social distancing measures appear effective, mainly when implemented in conjunction with the isolation of people who test positive for COVID-19 and quarantining of anyone who has been in contact with them [31,37]. Our data indicates that that preventive behavior among the population of Ijuí, related to the SDA recommendation and DPR, did not reach 70% participation in social distancing in any of the five waves of the survey. We mentioned the threshold of 70% since it was proposed early that maintaining social distancing at a maximum of 76% could prevent 90,000 COVID-19-related deaths and keep intensive care units in São Paulo from being overwhelmed [38]. As of November 23, 2020, São Paulo has registered more than 394,000 COVID-19 cases and 14,000 deaths [7], suggesting that maintaining and strengthening current social distancing measures, isolating COVID cases, and quarantining people who have been in contact with others who have tested positive, is absolutely vital to avoid serious stress to Brazil’s healthcare system. Furthermore, it has been suggested that more restrictive recommendations can be more effective in reducing the number of infected subjects [1,38–41], and it is necessary to apply such recommendations immediately and rigorously, especially to control the spread of COVID-19 in schools, since children and teenagers may have a disproportional contribution to an increase in the transmission rates [18]. However, as reported by WHO, many countries have reported an increase in “pandemic fatigue” among the population, characterized by lack of motivation to follow the recommended social distancing behaviors to protect themselves and others from the virus (WHO) [42]. On March 13, 2020 the United Stated issued a national proclamation that almost immediately resulted in a large number of people sheltering at home and reducing their daily movements, in line with the MSDI trends we observed in Ijuí in March. From early April to mid-April, the MSDI reached an upper limit followed by a plateau, indicating “social distancing inertia” in Ijuí. After that, a reduction in social distancing measures occurred even in the states that maintained the recommendation of mobility restriction, an example of “quarantine fatigue” [17]. 

Q18. Statistical and epidemiological analyzes are appropriate and well described;

Answer 18. Thank very much for this comment. We hope that it can be useful for readers.

Q19. The work, in my view, constitutes an important study to understand the pandemic spread.

 Answer 19. Thank very much for this comment. We hope that it can be useful for readers.

References included in the new version of the manuscript:

7. Brazilian Ministery of Health. Brazilian national government COVID-19 website. 2020 [cited 6 Aug 2020]. Available: https://covid.saude.gov.br/

10. Chakraborty R, Parvez S. COVID-19: An overview of the current pharmacological interventions, vaccines, and clinical trials. Biochemical Pharmacology. 2020. doi:10.1016/j.bcp.2020.114184

11. Ng OT, Marimuthu K, Koh V, Pang J, Linn KZ, Sun J, et al. SARS-CoV-2 seroprevalence and transmission risk factors among high-risk close contacts: a retrospective cohort study. Lancet Infect Dis. 2020. 

12. Aleta A, Martín-Corral D, Pastore y Piontti A, Ajelli M, Litvinova M, Chinazzi M, et al. Modelling the impact of testing, contact tracing and household quarantine on second waves of COVID-19. Nat Hum Behav. 2020. doi:10.1038/s41562-020-0931-9

13. Block P, Hoffman M, Raabe IJ, Dowd JB, Rahal C, Kashyap R, et al. Social network-based distancing strategies to flatten the COVID-19 curve in a post-lockdown world. Nat Hum Behav. 2020. doi:10.1038/s41562-020-0898-6

14. Atalan A. Is the lockdown important to prevent the COVID-9 pandemic? Effects on psychology, environment and economy-perspective. Ann Med Surg. 2020. doi:10.1016/j.amsu.2020.06.010

15. Li Y, Campbell H, Kulkarni D, Harpur A, Nundy M, Wang X, et al. The temporal association of introducing and lifting non-pharmaceutical interventions with the time-varying reproduction number (R) of SARS-CoV-2: a modelling study across 131 countries. Lancet Infect Dis. 2020. doi:10.1016/s1473-3099(20)30785-4

16. Jefferies S, French N, Gilkison C, Graham G, Hope V, Marshall J, et al. COVID-19 in New Zealand and the impact of the national response: a descriptive epidemiological study. Lancet Public Heal. 2020. doi:10.1016/S2468-2667(20)30225-5

17. Zhao J, Lee M, Ghader S, Younes H, Darzi A, Xiong C, et al. Quarantine Fatigue: first-ever decrease in social distancing measures after the COVID-19 outbreak before reopening United States. 2020 [cited 15 Nov 2020]. Available: https://arxiv.org/abs/2006.03716

18. Taylor S. The psychology of pandemics: Preparing for the next global outbreak of infectious disease. Newcastle upon Tyne: Cambridge Scholars Publishing. Cambridge Sch. 2019.

34. Ajzenman N, Cavalcanti T, Da Mata D. More Than Words: Leaders’ Speech and Risky Behavior during a Pandemic. SSRN Electron J. 2020. doi:10.2139/ssrn.3582908

42. WHO. WHO/Europe discusses how to deal with pandemic fatigue. In: WHO/Europe discusses how to deal with pandemic fatigue [Internet]. 2020 [cited 6 Nov 2020]. Available: https://www.who.int/news-room/feature-stories/detail/who-europe-discusses-how-to-deal-with-pandemic-fatigue

43. Maglio SJ. An agenda for psychological distance apart from construal level. Soc Personal Psychol Compass. 2020. doi:10.1111/spc3.12552

44. Trope Y, Liberman N. Construal-Level Theory of Psychological Distance. Psychol Rev. 2010. doi:10.1037/a0018963

45. Slovic P. The Psychology of Risk [Psicologia do Risco]. Saude Soc Sao Paulo. 2010.

Looking forward to hearing from PlosOne soon, 

Yours faithfully,

Prof. Dr. Thiago Gomes Heck

Coordinator of Post Graduate Program in Integral Attention to Health

Regional University of Northwestern Rio Grande do Sul State (UNIJUI)

Ijuí, RS, Brazil

---

## [Decision Letter · Decision Letter 1]

21 Jan 2021

Insufficient social distancing may contribute to COVID-19 outbreak: the case of Ijuí city in Brazil

PONE-D-20-19438R1

Dear Dr. Heck,

We’re pleased to inform you that your manuscript has been judged scientifically suitable for publication and will be formally accepted for publication once it meets all outstanding technical requirements.

Kind regards,

Amir H. Pakpour, Ph.D.

Academic Editor

PLOS ONE

Additional Editor Comments (optional):

Reviewers' comments:

Reviewer's Responses to Questions

**Comments to the Author**

1. If the authors have adequately addressed your comments raised in a previous round of review and you feel that this manuscript is now acceptable for publication, you may indicate that here to bypass the “Comments to the Author” section, enter your conflict of interest statement in the “Confidential to Editor” section, and submit your "Accept" recommendation.

Reviewer #1: All comments have been addressed

2. Is the manuscript technically sound, and do the data support the conclusions?

Reviewer #1: Yes

3. Has the statistical analysis been performed appropriately and rigorously? 

Reviewer #1: Yes

4. Have the authors made all data underlying the findings in their manuscript fully available?

Reviewer #1: Yes

5. Is the manuscript presented in an intelligible fashion and written in standard English?

Reviewer #1: Yes

6. Review Comments to the Author

Reviewer #1: I read the manuscript reviewed by the authors very carefully. I could see the care, the attention, the care with which the authors reviewed the manuscript. They responded, essentially and in depth, to all the suggestions and comments we made during our first review. Importantly, the authors were very humble and didactic in their responses. I was really very pleased to participate in this review process in which, for me, there was scientific respect for peers and the ethics that must prevail in the interaction between researchers.

The manuscript is very enriched, presents a lot of details and faithfully portrays the scenario of distance (social, physical) in a large city in Brazil

The authors should be congratulated for the excellent work and review.

Te best wishes

7. PLOS authors have the option to publish the peer review history of their article (what does this mean?). If published, this will include your full peer review and any attached files.

Reviewer #1: **Yes: **José Aparecido da Silva, Full Professor

---

## [Editor Report · Acceptance letter]

26 Jan 2021

PONE-D-20-19438R1 

Insufficient social distancing may contribute to COVID-19 outbreak: the case of Ijuí city in Brazil 

Dear Dr. Heck:

I'm pleased to inform you that your manuscript has been deemed suitable for publication in PLOS ONE. Congratulations! Your manuscript is now with our production department. 

Kind regards, 

on behalf of

Dr. Amir H. Pakpour 

Academic Editor

PLOS ONE